# The *Gastrodia elata* genome provides insights into plant adaptation to heterotrophy

Yuan Yuan[1], Xiaohua Jin[2], Juan Liu[1], Xing Zhao[3], Junhui Zhou[1], Xin Wang[1], Deyi Wang[2], Changjiangsheng Lai[1], Wei Xu[3], Jingwen Huang[1], Liangping Zha[4], Dahui Liu[5], Xiao Ma[2], Li Wang[6], Menyan Zhou[3], Zhi Jiang[3], Hubiao Meng[1], Huasheng Peng[4], Yuting Liang[1], Ruiqiang Li[3], Chao Jiang[1], Yuyang Zhao[1], Tiegui Nan[1], Yan Jin[1], Zhilai Zhan[1], Jian Yang[1], Wenkai Jiang 🄳 [3] & Luqi Huang[1]

We present the 1.06 Gb sequenced genome of *Gastrodia elata*, an obligate mycoheterotrophic plant, which contains 18,969 protein-coding genes. Many genes conserved in other plant species have been deleted from the *G. elata* genome, including most of those for photosynthesis. Additional evidence of the influence of genome plasticity in the adaptation of this mycoheterotrophic lifestyle is evident in the large number of gene families that are expanded in *G. elata*, including glycoside hydrolases and urease that likely facilitate the digestion of hyphae are expanded, as are genes associated with strigolactone signaling, and ATPases that may contribute to the atypical energy metabolism. We also find that the plastid genome of *G. elata* is markedly smaller than that of green plant species while its mitochondrial genome is one of the largest observed to date. Our report establishes a foundation for studying adaptation to a mycoheterotrophic lifestyle.

[1] National Resource Center for Chinese Meteria Medica, Chinese Academy of Chinese Medical Sciences, 100700 Beijing, China. [2] Institute of Botany, Chinese Academy of Sciences (IBCAS), 100093 Beijing, China. [3] Novogene Bioinformatics Institute, 100083 Beijing, China. [4] Anhui University of Chinese Medicine, 230012 Hefei, China. [5] Hubei University of Chinese Medicine, 430065 Wuhan, China. [6] Institute of Medicinal Botany, Yunnan Academy of Agricultural Sciences, 650223 Kunming, China. These authors contributed equally: Yuan Yuan, Xiaohua Jin, Juan Liu, Xing Zhao, Junhui Zhou. Correspondence and requests for materials should be addressed to Y.Y. (email: y_yuan0732@163.com) or to W.J. (email: jiangwenkai@novogene.com) or to L.H. (email: huangluqi01@126.com)

Symbiotic associations between plants and fungi (mycorrhizae) began about 450 million years ago[1]. Most mycorrhizal associations are mutualistic, such that the host plant and mycorrhizal fungi exchange nutrients with each other[2]. However, mycoheterotrophs have evolved a special type of plant–fungi symbiosis in which a plant gets fixed carbon and other nutrients from fungal partners, rather than from photosynthesis[3]. One of the most interesting characteristics of orchids is the reliance on fungi for seed germination and nutrient absorption, for example, through formation of mycorrhiza with fungi. Over 99% of orchids show partial mycoheterotrophy in which young plants obtain carbon (C) nutrients from fungi prior to the development of green leaves, while adult plants are autotrophic. The extreme type of mycoheterotrophy in orchids is obligate mycoheterotrophy, in which plants are achlorophyllous (lack chlorophyll) throughout their life cycle and therefore fully dependent on fungi for nutrition.

*Gastrodia elata* (Orchidaceae) is an orchid popularly used in traditional Chinese medicine that has a fully mycoheterotrophic lifestyle with highly reduced leaves and bracts in scape, although field guides and systematists often refer to the plants as leafless[4,5]. During its life cycle, in associates with at least two types of fungi: *Mycena* for seed germination and *Armillaria mellea* for plant growth. To obtain nutrition, it forms an association with *A. mellea*[6] and more than 80% of its ~36-month lifespan is spent underground as a tuber (Fig. 1a). These features of the plant are putative adaptions to its obligate mycoheterotrophic lifestyle. *G. elata* thus offers the possibility of obtaining a valuable insight into the genetic basis of mycoheterotrophy. Here we present a high-quality reference genome assembly of *G. elata* (Orchidaceae), and use it to investigate the molecular basis of its full mycoheterotrophic life cycle. The observations presented here will be of value for functional ecological studies seeking to understand the mechanisms and evolutionary basis of plant–fungal associations.

## Results

**Sequencing and annotation**. The genome of a *G. elata* individual was sequenced using a whole-genome shotgun (WGS) approach (Supplementary Table 1). Through K-mer distribution analysis, the genome size was estimated to be 1.18 Gb (Supplementary Fig. 1). The assembly consisted of 3779 scaffolds, with a scaffold N50 of 4.9 Mb (total length = 1061.09 Mb) and contig N50 of 68.9 kb (total length = 1025.5 Mb) (Supplementary Table 2). Overall, 98.51% of the raw sequence reads could be mapped to the assembly, suggesting that our assembly results contained comprehensive genomic information (Supplementary Table 3). Gene region completeness was evaluated by RNA-Seq data (Supplementary Table 4): of the 80,646 transcripts assembled by Trinity, 98.66% could be mapped to our genome assembly, and 94.41% were considered as complete (more than 90% of the transcript could be aligned to one continuous scaffold). The completeness of gene regions was further assessed using CEGMA (conserved core eukaryotic gene mapping approach): 239 of 248 (96.37%) conserved core eukaryotic genes from CEGMA were captured in our assembly, and 217 (87.5%) of these were complete (Supplementary Table 5).

Much of the *G. elata* genome (66.18%) was occupied by transposable elements (TEs). Class I (retrotransposons) and Class II (DNA transposons) TEs accounted for 55.94% and 4.38% of the genome, respectively (Supplementary Table 6). Long terminal repeats (LTRs) formed the most abundant category of TE, with LTR/Gypsy and LTR/Copia occupying 45.04% and 7.10% of the genome, respectively (Supplementary Table 7). Global activity of LTRs was similar between *G. elata* and *Phalaenopsis equestris*, while *Dendrobium officinale* presented a recent burst of LTR

activities (Supplementary Fig. 2a). Compared to *P. equestris*, all LTR families in *G. elata* had fewer members, except a substantive expansion of del family (Supplementary Table 8 and Supplementary Fig. 2b). Through a combination of ab initio prediction, homology search, and RNA sequence-aided prediction, 18,969 protein-coding genes were predicted in the *G. elata* genome. Of these genes, 81.6% were functionally annotated (Supplementary Table 9) and 88.69% had detectable transcripts in an RNA-seq analysis of protocorms, tubers (juvenile, immature, and mature tubers), and scapes (Supplementary Table 10). Our transcriptomics analysis revealed that there were 10,548 differentially expressed genes among the five growth stages; these differentially expressed genes clustered into five distinct groups that were representative of the particular stages of growth of *G. elata* (Supplementary Fig. 3, Supplementary Table 11 and Supplementary Note).

**Phylogeny and whole-genome duplication**. Comparison of the sequenced genomes of the orchid species *G. elata*, *P. equestris*[7], and *D. officinale*[8,9] indicated that they diverged approximately 67 million years ago (Fig. 1b and Supplementary Fig. 4). Two ancient whole-genome duplication (WGD) events are evident in the *G. elata* genome; these events can also be discerned in the genomes of *P. equestris* and *D. officinale* suggesting they occurred prior to the divergence of the three orchid species (Supplementary Fig. 5). The older WGD event might represent the heWGD event[10] shared by most monocots, while the younger WGD event were likely shared by all extant orchids and might contribute to the divergence of orchid, as suggested in *Apostasia shenzhenica* genome[11].

**Extensive gene lost in *G. elata* genome**. Compared to *P. equestris* (29,431 protein-coding genes) and *D. officinale* (28,910 protein-coding genes), *G. elata* has a relative small proteome size (18,969 protein-coding genes). The estimated proteome size of *G. elata* is the smallest theoretical proteome so far identified among angiosperm genomes (Supplementary Table 12). Comparison of *G. elata*, *P. equestris*, and *D. officinale* genes that have functional annotation information revealed global gene set reduction in the *G. elata* genome. For example, almost all second level gene ontology (GO) categories had fewer genes in *G. elata* than in the other two species, and 9 of these categories (16.7%) were significantly reduced (Fisher's Exact test, $p < 0.05$, Supplementary Fig. 6 and Supplementary Table 13). We also found that several Pfam domain families were significantly reduced in the *G. elata* genome (Supplementary Table 14). Among the 14 angiosperm used in the phylogenetic analysis, *G. elata* had the lowest number of gene families; moreover, *G. elata* had on average the lowest number of genes in each gene family (Fig. 1c, and Supplementary Table 15). This consistently low number of genes and gene families suggests that many gene families have been eliminated from the *G. elata* genome, and further suggests that many of the remaining gene families have contracted. Gene family expansion and contraction analysis based on maximum likelihood modeling of gene gain and loss confirmed that 3586 gene families had undergone contraction in *G. elata*, much more compared to the other two orchid genomes (Supplementary Fig. 7 and Supplementary Table 16). A Benchmarking Universal Single-Copy Orthologs (BUSCO) analysis, which assessed 956 orthologous groups with genes present as single-copy in at least 90% of plant genomes[12], revealed that 195 (20.4%) highly-conserved genes were missing from the *G. elata* genome. This rate of absence is much higher than in the genomes of the 13 land species that were included in this analysis (Supplementary Table 17). All of these analyses indicate that *G. elata* has undergone extensive gene

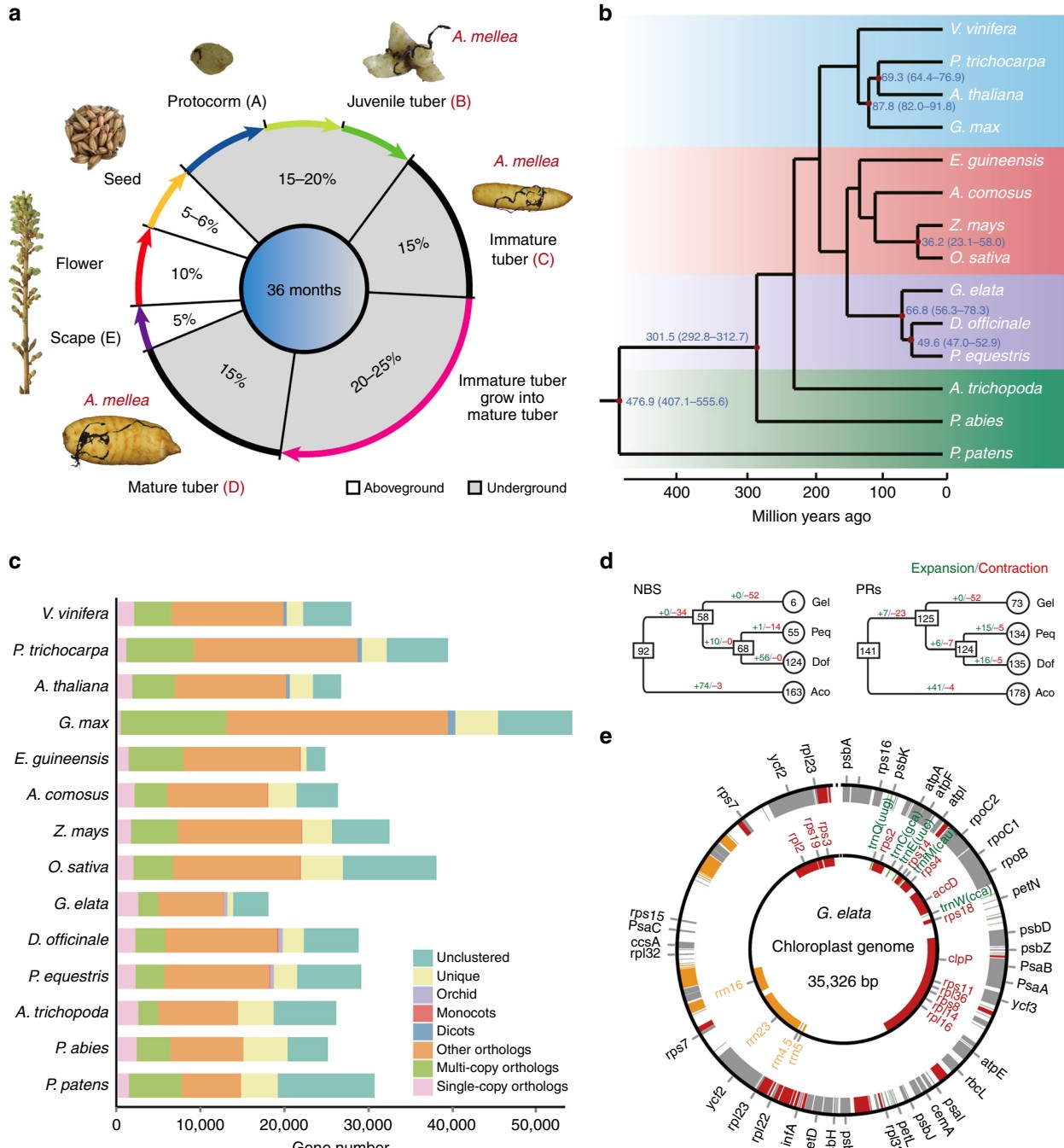

**Fig. 1** *Gastrodia elata* life cycle and gene-family contraction. **a** The main developmental stages of *G. elata*. Seeds develop into a protocorm stage without requiring *A. mellea* (A). The protocorm then differentiates into a corm stage after commencing its association with *A. mellea*; note that lateral buds develop into juvenile tubers (B). Young and immature tubers (C). Mature tuber with an emergent young scape (D). Scape (stem and inflorescence) of mature plants (E). **b** Phylogenetic tree of 14 plant species including *G. elata*. The red dot represents a calibration point determined from the timetree website. **c** Bar graph of the number of protein-coding genes in each of the species. Single-copy orthologs include common orthologs with one copy in specific species. Multi-copy orthologs include common orthologs with multiple copy numbers in specific species. Other orthologs include genes from families shared in 2–13 species. 'Eudicot' clusters with eudicots. 'Monocots' clusters with monocotyledonous plants. 'Orichd' clusters with *G. elata*, *P. equestris*, and *D. officinale*. 'Unclustered' include genes that cannot be clustered into gene families. **d** Analysis of gene numbers in the genomes of four species for the nucleotide-binding site gene family (NBS), the pathogenesis-related protein (PR) family (Gel *G. elata,* Peq *P. equestris*, Dof *D. officinale*, Aco *A. comosus*). Numbers in circles represent the number of family members in each genome, and numbers with plus or minus signs indicate, respectively, the number of duplicated or deleted genes. **e** The plastid genomes of *P. equestris* (outer circle) and *G. elata* (inner circle). Red, protein; orange, rRNA; green, tRNA; gray, genes lost from the plastid genome of *G. elata*

losses, even for genes that were conserved in other plant species that have also undergone extensive lost events.

The absence of these genes is unlikely to be due to genome assembly problems because 98.66% of the transcripts assembled from transcriptome data could be mapped to the assembly. Another possibility is that several genes were missed due to gene prediction problems. By mapping RNA reads onto the annotated genome, we found that the majority of RNA reads (>86%) from all *G. elata* tissues could be mapped to annotated exon regions (Supplementary Table 18). This rate of mapping was comparable to that achieved in the well-annotated rice genome and higher

than in the *P. equestris* genome (Supplementary Table 18). Through analysis of gene synteny among *G. elata* and *P. equestris* and *D. officinale*, we detected 2961 gene deletion events in *G. elata* versus *P. equestris*, and 3120 gene deletion events in *G. elata* versus *D. officinale* (Supplementary Table 19). Further TBLASTN searches of these deleted genes recovered less than 3% of them. Of these genes, fewer than 15% were supported by RNA-seq data (Supplementary Table 19). Both the RNA mapping results and the synteny deletion analysis confirmed that our gene prediction was comprehensive; thus, the possibility of missing gene annotations was low. Finally, PCR amplification of 18 lost genes

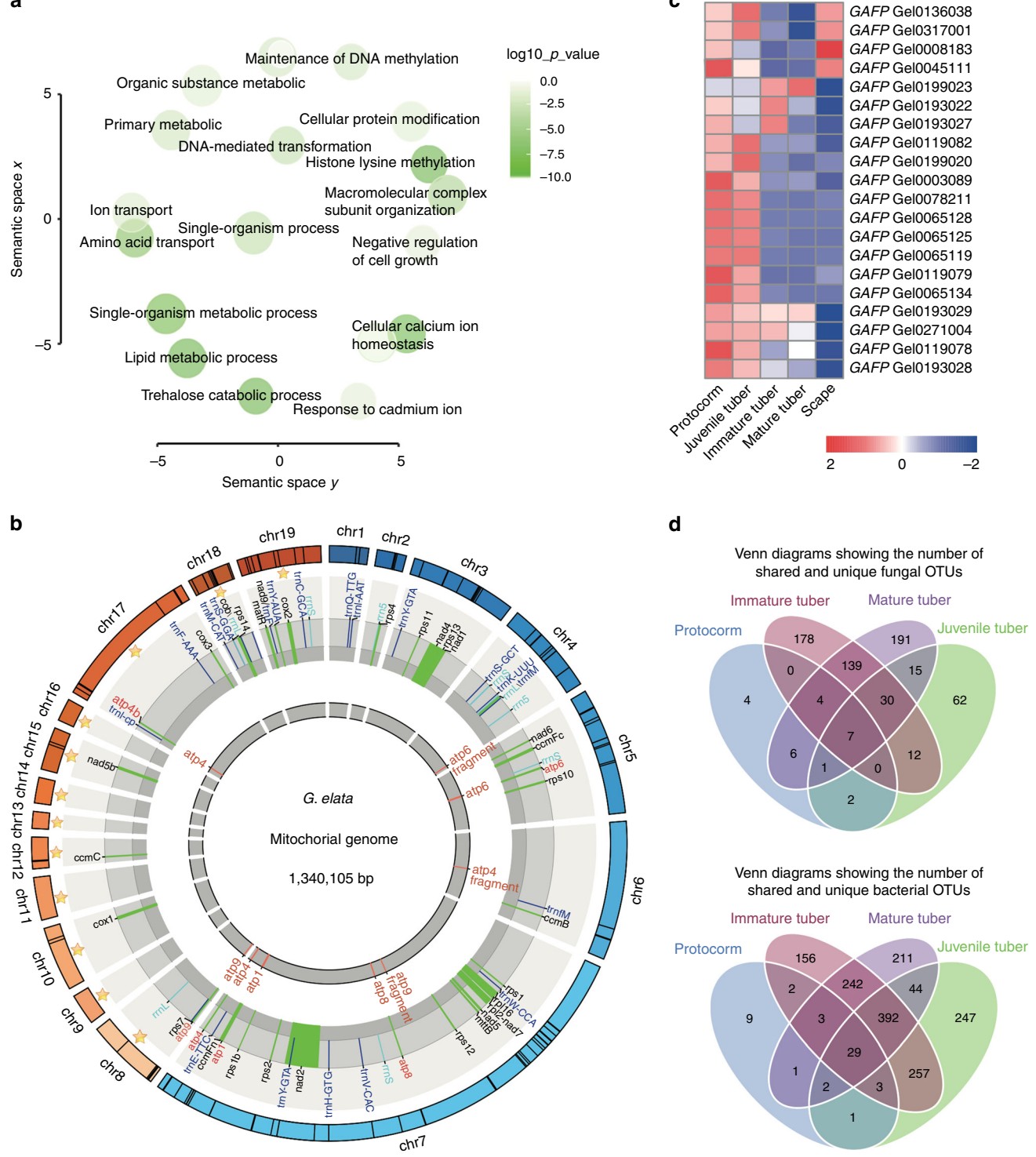

(*atp*D, *atp*G, *Ihc*A, *Ihc*B, *psa*D, *psa*F, *psa*L, *psa*N, *psb*O, *psb*R, *psb*Y, *psb*27, *psb*28, *pet*C, *pet*E, *ICS*, *DHAR*, and *TRX*) confirmed that all were absent from the *G. elata* genome (Supplementary Figs. 8, 9 and Supplementary Table 20). Thus, the global gene losses in *G. elata* represent evolutionary events, and might be the result of adaption to an obligate mycoheterotrophic lifestyle.

Both pseudogenizations and genome rearrangements contributed to the gene lost process of *G. elata*. We found 876 and 1080 pseudogenes in *G. elata* using *P. equestris* and *D. officinale* genes as seeds, respectively (Supplementary Tables 21–24). Through a whole-genome alignment between *G. elata* and *P. equestris*, we found 487 genes were lost due to local rearrangements (SV genes, Supplementary Tables 25 and 26). Functional genes in *G. elata* were located closer to transposable elements than to pseudogenes and SV genes, suggesting that transposable element did not play significant role during the gene lost processes in *G. elata* (Supplementary Fig. 10). Thus the gene lost processes in *G. elata* might be dominated by random mutations as we found many pseudogenes in *G. elata*. Notably, compared with the genomes of *P. equestris*, *D. officinale*, *A. comosus* (pineapple, used in this study as an outgroup) and *Arabidopsis thaliana* (used in this study as an outgroup), the *G. elata* genome has a reduced number of genes related to plant resistance to pathogens, such as the NBS (the nucleotide-binding site) gene family, PR (pathogenesis-related) gene family, and genes of antioxidant proteins (Fig. 1d, Supplementary Fig. 11 and Supplementary Table 27). For example, the *ICS* gene, which is known to function as a primary modulator of salicylic acid-based plant defense responses[13], is absent from the *G. elata* genome, illustrating the loss of a key gene for systemic acquired resistance (Supplementary Figs. 8, 9 and Supplementary Table 27).

**Degeneration of photosynthesis system**. As *G. elata* does not perform photosynthesis, it was unsurprising that genes enriched for 'chloroplast' and 'plastid' annotations were strongly represented among the missing genes (Supplementary Table 28). To further investigate the putative functions of missing genes, we examined genes related to the photosynthetic apparatus, namely Photosystem I, Photosystem II, Cytochrome b$_6$f, Cytochrome C$_6$, ATP synthase, and Rubisco[14]. Of the 35 nuclear genes coding for photosynthetic apparatus proteins (NEP), only 12 were present in the *G. elata* genome; this is significantly fewer than in *A. thaliana*, *A. comosus*, *P. equestris*, and *D. officinale* (Supplementary Tables 28, 29). We assume that these genes were non-functional because their full complements of subunits were not present.

We also sequenced and assembled the plastid genome of *G. elata*. We found that the plastid genome of *G. elata* (35,326 bp) was dramatically restructured and reduced in size, in a similar manner to the reduction in gene number observed for the nuclear genome (Fig. 1e), compared to the plastid genomes of *P. equestris* (148,958 bp)[15] and *D. officinale* (152,221 bp)[16], the two other

orchid species with sequenced genomes. The plastid genomes of these two species comprise two single-copy regions (a large and a small single-copy region) and the two identical large inverted repeats (IRs) encode 75 and 76 genes, respectively, that are most associated with photosynthesis. The *G. elata* plastid genome has lost one IR and encodes only 19 protein-coding genes (Fig. 1e), suggesting that *G. elata* is an ancient mycoheterotroph and that its plastid genome is in the last stage of a 'degradation ratchet', i.e., retention and loss of the five core nonbioenergetic genes[17,18]. Excluding the possibility that these genes were missed by our genome assembly, the transcriptome sequencing analysis indicated that none of the deleted plastid or nuclear encoded genes were expressed in *G. elata*, while the five core nonbioenergetic genes, *trnE*, *aacD*, *clpP*, *ycf1*, and *ycf2*, were moderately to highly expressed in all five stages in *G. elata* (Supplementary Tables 30, 31). These results clearly show that both the plastid and nuclear genomes of *G. elata* have lost most of the genes required for photosynthesis, although the highly degraded plastome is still essential for this full mycoheterotroph.

**Expansion of mitochondrial genome**. Although the *G. elata* genome has clearly undergone extensive gene loss, we found that 430 gene families (19 by a significant margin), containing 1532 genes (184 by a significant margin), showed expansion in *G. elata* compared to *P. equestris*, *D. officinale*, and *A. comosus* (Supplementary Fig. 7 and Supplementary Tables 32, 33). These genes are enriched for GO terms related to several metabolic processes (Fig. 2a, Supplementary Table 32). We speculate that these expanded genes are related in some way to the functional requirements of the obligate mycoheterotrophic lifestyle of *G. elata*. We first sequenced and assembled the mitochondrial genome to explore this idea, and the mitochondrial genome *G. elata* is markedly expanded in size (1339 kb, Fig. 2b) compared to the mitochondrial genomes of most other seed plants[19]. Thirty-seven protein-coding genes were annotated, and one subunit of mitochondrial ATP synthase, atp4, had two copies in the mitochondrial genome of *G. elata* and was highly expressed in the cortex layer (Supplementary Table 34). In addition, 36 of the genes had detectable expression in mature tubers (epidermis, cortex, and parenchymal cell) using a tissue-specific qPCR-based analysis (Supplementary Table 34).

**Management of symbiotic microbials**. We next explored how gene expansion in *G. elata* may have contributed to its association and interactions with fungal microbiota. The monocot mannose-binding lectin antifungal protein family (GAFP) of *G. elata* contains 20 genes, compared to only 3 in *A. comosus* and 0 in *A. thaliana* (Supplementary Table 27). GAFP proteins have been documented to inhibit the growth of both ascomycete and basidiomycete fungal plant pathogens in vitro[20]. More than 80% of the *GAFP* genes were highly expressed in protocorms and juvenile tubers, the growth stages that occur before *G. elata* establishes

**Fig. 2** Gene expansion in *G. elata* and microbial community analysis. **a** REVIGO semantic similarity scatter plot of Biology Process Gene Ontology terms for expanded genes in *G. elata*. In semantic spaces, the proximity between circles represents relatedness (similarity) of the GO terms. Similar GO terms are close together in the plot. The axes in the plot have no intrinsic meaning, but were used to measure pairwise similarities between GO terms. Color indicates degree of enrichment for each process presented as the *p*-value from the hyper-geometric test. **b** The draft mitochondrial genome of *G. elata*. Nineteen contigs are manually displayed as a circle, including 12 circular contigs in orange (ornamented with stars) and 7 linear contigs (in blue). The genes are indicated in the middle circle, and are color coded as follows: *trn* (blue), *rrn* (light blue), *atp* (red), and other protein-coding genes (black). The duplicated *atp* genes and their fragments are detailed in the inner circle. The duplicated genes are suffixed with 'b', and the gene fragments are suffixed with 'fragment'. **c** Gene expression heat map of the normalized RNA-Seq data for genes encoding the monocot mannose-binding lectin antifungal proteins (GAFP) in *G. elata*[16]. The units indicate the expression levels of different gene members of *GAFP* in the protocorm, juvenile tuber, immature tuber, mature tuber, and scape of *G. elata* (only shown where the gene expression level RPKM > 1, *n* = 3). **d** Venn diagrams showing the number of shared and unique fungal and bacterial operational taxonomic units (OTUs) based on the ITS and 16S sequence analyses in protocorms, juvenile tubers, immature tubers, and mature tubers of *G. elata*. OTUs showed the composition and abundance of the microbe species, which were defined at 3% dissimilarity

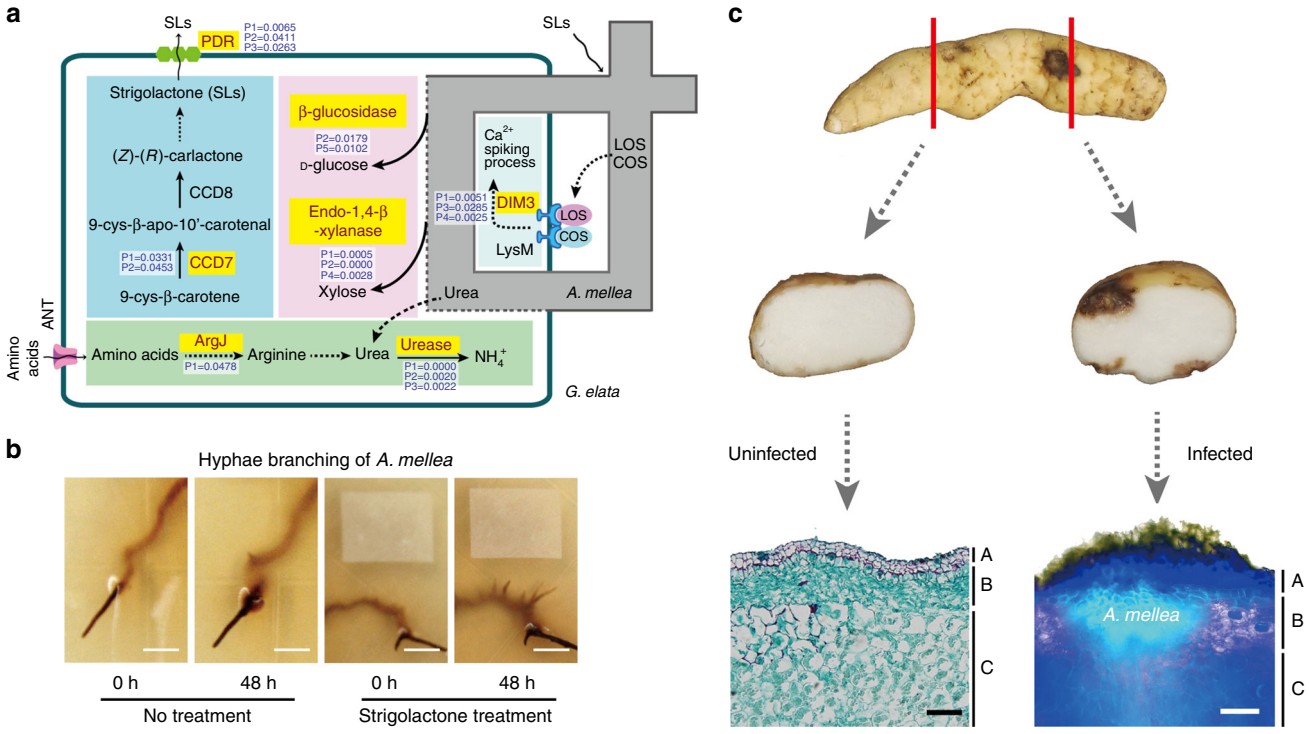

**Fig. 3** Strigolactone as a putative signal compound in *G. elata* in its mycoheterotrophic symbiotic relationship with *A. mellea*. **a** Overview of proposed signaling and nutrition transfer in *G. elata*. The red-labeled genes are expanded in the *G. elata* genome and *P*-value of Fisher's exact test of gene number <0.05 (Supplementary Tables 38, 40, 41). ANT, ANT1-like aromatic and neutral amino acid transporters; ArgJ, glutamate N-acetyltransferase; CCD, carotenoid cleavage dioxygenases; COS, chitooligosaccharides; DMI3, does-not-make-infections 3 subfamily; LOS, lipochitooligosaccharides; LysM, LysM-receptor-like kinases; PDR, ABC transporter. P1, *p*-value of Fisher's exact test of gene number in *G. elata* genome compared to *P. equestris*, *D. officinale*, *A. comosus*, and *A. thaliana*; P2, *p*-value of Fisher's exact test of gene number in *G. elata* genome compared to *A. thaliana*; P3, *p*-value of Fisher's exact test of gene number in *G. elata* genome compared to *A. comosus*; P5, *p*-value of Fisher's exact test of gene number in *G. elata*, *P. equestris*, and *D. officinale* genome compared to *A. comosus* and *A. thaliana*. **b** Branching of *A. mellea* hyphae was significantly promoted after strigolactone treatment (Supplementary Fig. 13). Scale bar was 5 mm. **c** Cross sections and micrographs of immature *G. elata* tubers in association with *A. mellea* (A, epidermis; B, cortex; C, inner parenchyma cells, Supplementary Fig. 14). The black scale bar on the left was 100 μm and the white one on the right was 20 μm

a stable symbiotic association with *A. mellea* (Fig. 2c). 4-Hydroxybenzyl alcohol (*p*-PA), the precursor of the phytoalexin gastrodin is a major phenolic compound of *G. elata*[21]. The expression of *p*-PA biosynthesis genes (e.g., cinnamate 4-hydroxylase, *C4H*, alcohol dehydrogenase, *ADH*, hydroxybenzaldehyde synthase, *HBS*)[21] was relatively high in protocorms and juvenile tubers. Ultra performance liquid chromatography coupled with quadrupole time-of-flight mass spectrometry (UPLC-ESI-Q-TOF-MS) metabolite analysis revealed that *p*-HA is present in *G. elata* tubers but not in *A. mellea* hyphae (sampled from tuber, wood, and PDA medium). S-(*p*-HA)-glutathione was detected in both *G. elata* tubers and in *A. mellea* hyphae sampled from tubers (Supplementary Fig. 12, Supplementary Table 35), which putatively suggests that *G. elata* may transport this phytoalexin to *A. mellea* and prevent the excessive growth of *A. mellea*. To investigate the effect of *A. mellea* on microbial management in *G. elata*, we performed a 16S ribosomal (rRNA) and rDNA ITS sequencing analysis and found that the diversity of bacterial and microbial species was significantly lower during the protocorm stage than at other growth stages (*p* < 0.05), which was consistent with the pattern of gene expression of *GAFP* (Fig. 2c, d, Supplementary Tables 36, 37). This increased diversity of bacteria and fungi during the juvenile tuber to mature tuber periods implies that a compatible mycorrhizal fungus (*A. mellea*) can affect the structure of the microbial community associated with its host and greatly reduce the antifungal and antibacterial activities as a symbiotic association with *A. mellea* is established.

**Signaling and nutrition transfer in *G. elata***. Without the ability to perform photosynthesis, *G. elata* depends completely on its symbiotic fungus for nutrition. It is thus obvious that the signaling pathways related to the establishment of this symbiotic relationship are crucial for *G. elata*. Some of the mechanisms underlying the symbiotic interaction between *G. elata* and *A. mellea* are similar to those for interactions between other plants and arbuscular mycorrhizal (AM) fungi[22]. The *G. elata* genome contains many of the genes known to participate in AM associations (Fig. 3a, Supplementary Table 38). Key genes for biosynthesis and secretion of strigolactone were expanded in *G. elata* (e.g., carotenoid cleavage dioxygenases, CCDs, for biosynthesis[23] and ABC transporters, PDRs, for secretion[24]) (Supplementary Table 38). It is known that strigolactone can stimulate hyphal branching and development of arbuscular mycorrhizal fungi, which increases the chances of an encounter with a host plant[24]. We conducted growth assays and confirmed that strigolactone had similar branch-inducing effects in *A. mellea* (Fig. 3b, Supplemental Fig. 13). The expanded number of genes encoding CCDs and PDRs suggests that *G. elata* has enhanced its ability to interact with *A. mellea* to increase the efficiency of the establishment of the symbiotic relationship essential for its nutrition and metabolism. Calmodulin-dependent protein kinase genes of the *does-not-make-infections 3* subfamily (*DMI3*) were also doubled or tripled in *G. elata* (10 genes) compared to *P. equestris* (3 genes), *D. officinale* (5 genes), and *A. comosus* (4 genes); these genes participate in the $Ca^{2+}$ spiking process that has been shown to regulate the colonization of plants by fungi[25].

After *A. mellea* colonizes *G. elata*, fungal growth is restricted to its cortex layer (Fig. 3c, Supplemental Fig. 14). We performed a tissue-specific qPCR-based analysis of 10 genes in *G. elata* tubers and found that PDR transcripts, which mediate secretion of strigolactone to the extracellular space, were highly abundant in the cortex layer (Supplementary Fig. 15). This finding suggests that *G. elata* may preferentially guide *A. mellea* to colonize its cortex layer. Similarly to ATP synthases, we found that some glycoside hydrolases from gene families that have expanded in the *G. elata* genome were also highly expressed in the cortex layer, supporting the idea that *A. mellea* hyphal walls are digested in the cortex layer of *G. elata* tubers (Supplementary Table 39). The expanded endo-β-1,4-D-xylanase and β-glucosidase may have become neofunctionalized to cleave fungal glycan substrates during the digestion of hyphal walls of *A. mellea* (Fig. 3a, Supplementary Table 40)[26,27].

Given that the ANT1-like aromatic and neutral amino acid transporters (ANT)[28] are known to translocate arginine (Arg), which is a key component in nitrogen translocation in arbuscular mycorrhizal fungi[29], it seems likely that Arg in *G. elata* is related to mycoheterotrophic symbiosis (Fig. 3a, Supplementary Table 41). It is known that arginases can hydrolyze Arg into urea in mycelia, which is further hydrolyzed to ammonium and carbonic acid by ureases[30]. Although *P. equestris*, *D. officinale*, *A. comosus* and *A. thaliana* have only one copy of glutamate N-acetyltransferase (ArgJ), an enzyme of the arginine biosynthesis pathway[31], *G. elata* has three copies (Supplementary Fig. 11 and Supplementary Table 41). The number of genes encoding ureases is drastically expanded in *G. elata* (9 genes) compared with *P. equestris* (2 genes), *D. officinale* (2 genes), *A. comosus* (1 gene), and *A. thaliana* (1 gene) (Supplementary Table 41). This suggests that urea metabolism might be an important source of nitrogen for *G. elata* (Fig. 3a).

**Conclusion**. The extensive deletion and expansion of genes, especially the global reduction of gene complements in almost all functional categories in the *G. elata* genome, provides a powerful example of how a plant with a fully heterotrophic life cycle has made use of genome plasticity to achieve extensive neofunctionalization and gene loss. Our results establish a unique opportunity for researchers to understand how plants that have abandoned photosynthesis continue to persist and thrive.

## Methods

**Plant materials and DNA preparation**. The experimental materials of *Gastrodia elata* were harvested from Xiaocaoba in Yunnan Province (latitude 27.79°N longitude 104.24°E) located in the southwestern China. Genome sequencing and assembly was done on the scape of beige-scape *G. elata*. Five transcriptomes were sequenced from five different *G. elata* tissues (protocorm, juvenile tuber, immature tuber, mature tuber, scape). Four different *G. elata* tissues (protocorm, juvenile tuber, immature tuber, mature tuber) were collected to investigate the diversity of microbial communities. High-quality genomic DNA was extracted using the Qiagen DNeasy Plant Mini Kit.

**Genome sequencing and assembly**. Multiple paired-end and mate-pair libraries were constructed with a spanning size that ranged from 180 bp to 20 kb. Sequencing was conducted on an Illumina HiSeq 2500 platform. In total, 179.1 Gb raw sequencing reads were produced (Supplementary Table 1). Raw sequencing reads were subjected to filtering to remove (1) low quality reads with low quality bases (>50% bases with $Q$-value ≤8); (2) reads with Ns >10% of the read length; (3) reads with adapter contamination; and (4) duplicated reads caused by PCR during library construction. Filtered data were assembled using ALLpaths-LG (version 44080)[32], where overlapping paired-end reads with an insert size of 230 nucleotides were used as fragment libraries, and all other libraries (>230 nucleotide insert size) were used as jumping libraries. The Allpaths-LG assembly was run with default settings, then a gap filling step was carried out using GapCloser based on the paired-end information of the paired-end reads that had one end mapped to the unique contig and the others located in the gap region (http://sourceforge.net/projects/soapdenovo2/files/GapCloser).

**Genome-quality evaluation**. To evaluate the completeness of the assembly and the uniformity of the sequencing, all the paired-end reads were mapped to the assembly using BWA[33]. The mapping rate was 98.51% and the genome coverage was 99.84%. This result suggested that our assembly results contained almost all the information in the reads (Supplementary Table 3). Gene region completeness was evaluated from the scape tissue, of 80,646 transcripts assembled by Trinity[34], 98.66% could be mapped to our genome assembly, and 94.41% were considered as complete (more than 90% of the transcript could be aligned to one continuous scaffold). CEGMA[35] (Core Eukaryotic Genes Mapping Approach) defined a set of conserved protein families that occur in a wide range of eukaryotes, and identified their exon–intron structures in a novel genomic sequence. Through mapping to the 248 core eukaryotic genes, a total of 239 genes with a ratio of 96.37% were found in *G. elata* (Supplementary Tables 5). Genome completeness was also assessed using BUSCO gene set analysis version 2.0[13] which includes a set of 956 single-copy orthologous genes specific to Plantae.

**Repetitive elements identification**. A combined strategy based on homology alignment and de novo search was used to identify repeat elements in the *G. elata* genome. For de novo prediction of transposable elements (TEs), we used RepeatModeler (http://www.repeatmasker.org/RepeatModeler.html), RepeatScout[36], and LTR-Finder[37] with default parameters. For alignment of homologous sequences to identify repeats in the assembled genome, we used RepeatProteinMask and RepeatMasker (http://www.repeatmasker.org) with the rebase library[38]. Transposable elements overlapping with the same type of repeats were integrated, while those with low scores were removed if they overlapped more than 80 percent of their lengths and belonged to different types (Supplementary Table 7).

**Dynamics of long terminal-repeat retrotransposons**. Intact Long terminal-repeat retrotransposons (LTR) were identified by searching the genomes of *G. elata*, *D. officinale* and *P. equestris* with LTRharvest[39] from Genome Tools v1.5.1. The candidate sequences were filtered by two-step procedure to reduce false positives. First, LTRdigest[40] was used to identify the primer binding site (PBS) motif based on the predicted tRNA sequences from tRNAscan-SE[41], and only elements contained PBS were retained; then protein domains (pol, gag and env) in candidate LTR retrotransposons were identified by searching against HMM profiles collected by Gypsy Databas (GyDB)[42]. Elements contained gag domain, protease domain, reverse transcriptase (RT) domain and integrase domain, which were considered as intact. Second, families of these intact LTR retrotransposons were clustered using the previously described method[43]. Finally, LTRs that did not contain protein domains or that belonged to families with less than 5 members were discarded. The EMBOSS program distmat[44] was used to estimate LTR divergence rates between the 5′- and 3′- LTR sequences of the intact LTRs (Supplementary Fig. 2).

**Gene prediction**. Gene prediction was conducted through a combination of homology-based prediction, ab initio prediction and transcriptome-based prediction methods. Protein repertoires of plants including *A. comosus*[10], *Amborella trichopoda*[45], *Arabidopsis thaliana* (phytozomev10), *Brachypodium distachyon* (phytozomev10), *D. officinale*[9], *O. sativa* (phytozomev10), *P. equestris*[7], *Vitus vinifera* (phytozomev10), *Sorghum bicolor* (phytozomev10) and *Zea mays* (phytozomev10) were downloaded and mapped to the *G. elata* genome using TBLASTN ($E$-value ≤ $1e^{-5}$). The BLAST hits were conjoined by Solar software[46]. GeneWise (version 2.4.1)[47] was used to predict the exact gene structure of the corresponding genomic region on each BLAST hit. Homology predictions were denoted as "Homology-set". RNA-seq data derived from protocorm, juvenile tuber, immature tuber, mature tuber, and scape (Fig. 1a) were assembled by Trinity (version 2.0)[41]. The Trinity assembly included 183,515 contigs with an average length of 592 bp. These assembled sequences were aligned against the *G. elata* genome by PASA (Program to Assemble Spliced Alignment)[48]. Valid transcript alignments were clustered based on genome mapping location and assembled into gene structures. Gene models created by PASA were denoted as PASA-T-set (PASA Trinity set). Besides, RNA-seq reads were directly mapped to the genome using Tophat (version 2.0.8)[49] to identify putative exon regions and splice junctions; Cufflinks (version 2.1.1) was then used to assemble the mapped reads into gene models (Cufflinks-set). Augustus (version 2.5.5)[50], GeneID (version)[51], GeneScan (version 1.0)[52], GlimmerHMM (version 3.0.1)[53], and SNAP (version)[54] were also used to predict coding regions in the repeat-masked genome. Of these, Augustus, SNAP and GlimmerHMM were trained by PASA-H-set gene models. Gene models generated from all the methods were integrated by EvidenceModeler (EVM)[48]. Weights for each type of evidence were set as follows: PASA-T-set > Homology-set > Cufflinks-set > Augustus > GeneID = SNAP = GlimmerHMM = GeneScan. The gene models were further updated by PASA2 to generate UTRs, alternative splicing variation information, which generated 26,872 gene models. Gene models only supported by ab initio evidence were filtered out. To reduce the possibility of missing and poorly annotated genes, we invested additional effort in annotating some gene families that could be missed by automated genome annotation, such as NBS-encoding genes. In total, 1943 protein sequences containing an NB-ARC domain were searched against the *G. elata* genome using TBLASN with a threshold of $1e^{-5}$. All

BLAST hits in the genome, together with 5000 bp flanking regions on both sides, were annotated by the GeneWise program. The resulting predictions were surveyed to verify whether they encoded NBS or LRR motifs using Pfam. We also focused on other genes, such as those related to photosynthesis, and transporter, and these were manually annotated through a combination of BLAST search and motif verification. Ultimately, a comprehensive non-redundant reference gene set was produced that contained 18,969 protein-coding gene models. Functional annotation of the protein-coding genes was carried out using BLASTP (E-value cut-off $1e-05$) against two integrated protein sequencing databases, SwissProt and TrEMBL[55]. Protein domains were annotated by searching against InterPro (Version 5.16)[56] and Pfam (Version 3.0) database[57], using InterProScan (version 4.8) and HMMER (version 3.1b1) (http://hmmer.janelia.org), respectively. The GO terms for genes were obtained from the corresponding InterPro or Pfam entry. The pathways in which the genes might be involved were assigned by BLAST against the KEGG databases (release 20150831)[58] with the E-value cut-off of $1e-05$.

**Identification of pseudogenes**. Pseudogenes in the G. elata genome were identified by searching against G. elata intergenic regions (using D. officinale or P. equestris protein sequences as the seed sequences (TBLASTN, E-value cut-off $1e-5$). Before the BLAST search, regions of the 18,969 true genes were masked. The BLAST hits were conjoined by Solar software. GeneWise was used to predict the pseudogene structures with the '-pseudo' parameter. Pseudogenes were then classified by PseudoPipe[59]. The PseudoPipe program applies a set of sequence identity and completeness cut-off to report a final set of good-quality pseudogene sequences. We used the following cutoffs: amino acid (AA) sequence identity >30% and match length >50 AA to filter out false positives. GeneWise results that fulfilled the cut-off criteria were denoted as high-confidence pseudogenes. High-confidence pseudogenes were then assigned to three categories. (1) Processed/retrotransposed pseudogenes (PSSDs), which formed through retrotransposition. Retrotransposition occurred by reintegration of a cDNA, a reverse transcribed mRNA transcript, into the genome at a new location. (2) Duplicated pseudogenes (DUPs), which formed through gene duplication, following by decay of genes, include frameshifts or premature stop codons. (3) Pseudogenic fragments (FRAGs), which were fragments that have high-sequence similarity to known proteins, but were too decayed to be reliably assessed as processed or duplicated. We used the following criteria to classify PSSDs, DUPs, and FRAGs: (i) PSSDs, exon number = 1, $0.7 <$ align ratio $\leq 0.95$, $0.3 \leq$ identity $\leq 0.95$; (ii) DUPs, exon number > 1, $0.3 \leq$ identity $\leq 0.95$, and existing insertion, deletion, termination, or frameshift; (iii) FRAGs, exon number = 1, align ratio < 0.7, $0.3 \leq$ identity $\leq 0.95$.

**Gene family construction**. Whole protein-coding gene repertoires from 14 plant genomes including G. elata, A. comosus [10], A. trichopoda [45], O. sativa (phytozome v10), Z. mays (phytozomev10), D. officinale[9], P. equestris[7], Elaeis oleifera[60], A. thaliana (phytozomev10), V. vinifera (phytozomev10), Populus trichocarpa (JGI), Glycine max (phytozomev10), Picea abies[61], Physcomitrella patens (ASM242v1) were used to construct a global gene family classification. To remove redundancy caused by alternative splicing variations, we retained only gene models at each gene locus that encoded the longest protein sequence. To exclude putative fragmented genes, genes encoding protein sequences shorter than 50 amino acids were filtered out. All-against-all BLASTp was employed to identity the similarities between filtered protein sequences in these species with an E-value cut-off of $1e^{-7}$. The OrthoMCL[62] method was used to cluster genes from these different species into gene families with the parameter of "-inflation 1.5".

**Phylogenetic tree reconstruction**. Protein sequences from 74 single-copy gene families were used for phylogenetic tree reconstruction. MUSCLE[63] was used to generate multiple sequence alignment for protein sequences in each single-copy family with default parameters. Then, the alignments of each family were concatenated to a super alignment matrix. The super alignment matrix was used for phylogenetic tree reconstruction through maximum likelihood (ML) methods. Before ML reconstruction, we used ProtTest[64] to select the best substitution models. The JTT + I + G + F model was selected as the best-fit model, and RAxMLwas used to reconstruct the phylogenetic tree[65].

**Species divergence time estimation**. Divergence time between 14 species was estimated using McMctree in PAML[66] with the options 'correlated molecular clock' and 'JC69' model. A Markov Chain Monte Carlo analysis was run for 20,000 generations, using a burn-in of 1000 iterations. Five calibration points were applied in the present study (Fig. 1): P. equestris and D. officinale divergence time (47~52.9 million years ago) [67], O. sativa and Z. mays divergence time (24–84 million years ago)[68,69], A. thaliana and P. trichocarpa divergence time (65–89 million years ago) [70,71], P. trichocarpa and G. max divergence time (56–89 million years ago)[56,57], and, root of land plants (407–557 million years ago) [57].

**Gene family expansion and contraction**. Expansion and contractions of orthologous gene families were determined using CAFÉ 2.2 (Computational Analysis of gene Family Evolution). The program uses a birth and death process to model gene gain and loss over a phylogeny. Large changes in gene family size in a phylogeny were tested by calculating p-values on each branch using the Viterbi method with a

randomly generated likelihood distribution. This method calculates exact p-values for transitions between the parent and child family sizes for all branches of the phylogenetic tree. Enrichment of Gene Ontology terms for G. elata expanded gene families were summarized and visualized using REVIGO (small list, similarity (0.5), SimRel similarity measure).

The expanded and contracted families focused on in this study were confirmed using Fisher's exact test. For each gene family, we compared the gene count of the tested family in G. elata (copy number of the tested family as numerator, total number of genes of the whole genome as denominator) versus the frequency in D. officinale[8,9], P. equestris[7], A. comosus[10], and Arabidopsis thaliana (phytozomev10). In addition, phylogenetic trees were constructed for each family to confirm gene gain or loss events. The extreme case of gene lost was that one gene was absent in the G. elata genome. To avoid false positive gene absence events caused by missing gene annotations, a TBLASTN search against the G. elata genome was carried out using protein sequences derived from other plant genomes.

**PCR verification of lost genes**. Total genomic DNAs were extracted by hexadecyl trimethyl ammonium bromide (CTAB) method from G. elata, A. thaliana, A. comosus, and D. officinale. PCR was carried out using SpeedSTAR™ HS DNA Polymerase (TaKaRa, Japan) and specific gene primers (Supplementary Table 20). PCR products were verified by agarose gel electrophoresis (Supplementary Figs. 8 and 9).

**Whole-genome duplication analysis**. A homolog search within the G. elata genome was performed using BLASTP (E-value $< 1e^{-7}$), and MCscanX was used to identify syntenic blocks within the genome. For each gene pair in a syntenic block, the 4DTv (transversion substitutions at fourfold degenerate sites) distance was calculated, and values of all gene pairs were plotted to identify putative whole-genome duplication events in G. elata.

**Inferring syntenic gene deletions in G. elata**. Proteins of G. elata, P. equestris and D. officinale were aligned using the BLASTp algorithm (E-value $< 1e^{-7}$). Alignments with matches of at least 30% identity and coverage higher than 30% were retained for comparison. The best reciprocal BLAST pairs between different genomes were extracted as putative orthologous gene pairs. Then, using gene location information in each species, we identified micro-synteny gene blocks between G. elata and the other two orchids. Putative gene loss events were traced from the synteny table using the flanking gene method. Given three genes A, B, and C in order, if gene A and C were presented as collinear orthologs in two genomes, but B was missed in one of the genome (for example, G. elata), then gene B was denoted as a possible lost gene in G. elata. To avoid false positives due to the failure of gene annotation, the G. elata intergenic genomic sequence between A and C was extracted, and a GeneWise prediction in this intergenic region was carried out using the B protein sequence from P. equestris as seed. If the predicted protein could be aligned to the seed protein with coverage >70%, and did not contain frameshift or premature stop codon mutations, this gene loss event was defined as a false positive and filtered out (Supplementary Table 19).

**Plastid genome**. Total genomic DNA was extracted using a modified CTAB from silica-dried tissues of G. elata (Supplementary Table 42). The DNA was sheared to 500 bp, and sequencing libraries were generated using the NEBNext Ultra DNA Library Prep Kit (according to the manufacturer's protocol) for sequencing on an Illumina Hiseq 2500 at the State Key Laboratory of Systematics and Evolutionary Botany, Chinese Academy of Sciences. The raw reads were filtered using NGSQCTOOLKIT v 2.3.3. The cleaned reads were mapped to Calanthe triplicata in GENEIOUS 9.0 (Biomatters, Inc., Auckland, New Zealand; http://www.geneious.com). Used reads were exported and assembled using SOAPDENOVO2. The plastid sequences were extracted from the total contigs using BLASTN 2.2.29+ and the C. triplicata plastome (GenBank ID: NC_024544) as subject sequence. The finished plastome scaffolds were reoriented according to the C. triplicata reference plastome. The boundaries of IRs were determined by BLAST, and finished manually. The plastomes of G. elata were determined using DOGMA with an e-value of 5, a 60% cut-off for protein-coding genes and 80% cut-off for tRNAs; the GENEIOUS annotation tool was used to determine the plastomes of C. triplicata and Oncidium Gower Ramsey (GenBank ID: NC_014056.1) as references. Linear plastome maps were drawn using OGDRAW.

**Mitochondrial genome**. Mitochondria were isolated from all G. elata tissues except the rhizome using previously described centrifugation methods[72]. A modified CTAB method was used to extract mt-DNAs[19]. The purified mt-DNAs were sequenced on an Illumina Hiseq 2500 to generate 100 bp paired-end reads at the State Key Lab of Systematic and Evolutionary Botany, Chinese Academy of Sciences (Beijing).

Eleven million raw reads were generated from sequencing and trimmed using Trimmomatic v0.35 to produce low quality reads. All sequenced plant mitochondrial genomes were downloaded from NCBI and used as a local blast database. To minimize the possibility of contaminated reads from plastid or nuclear genomes, filtered reads were first mapped to the local database and mapped reads were subsequently imported into Geneious v10.1.3 (Biomatters, Inc.,

Auckland, New Zealand; http://www.geneious.com) for initial assembly. The contigs generated by the initial assembly were used as seeds for further iterative mapping and extension processes. Velvet and Geneious were alternatively used during assembly with multiple combinations of k-mer lengths.

In most cases, the extension process of the assembly worked well. In particular, when the head and tail of a contig had an overlapping region and could not be further extended, this contig could be reasonably connected into a circle. Although several circles were produced during assembly, some problems did arise in the extension process. For example, some contigs were displayed as single lines because their boundaries were too difficult to determine due to poly tandoms or repeats in the mitogenome. The final assembled results were verified by remapping and some ambiguous regions with low coverage were further checked by PCR. Overall, 19 contigs with a total length of 1,340,105 bp were assembled including 12 circles (ranging from 13.5 to 120.6 kb) and 7 single lines (837,015 bp).

The assembled contigs were firstly annotated by NCBI-BlastN based on the local database with an $e$-value $< 1e^{-6}$. Then, the boundaries of each gene were confirmed by Mitofy and exported as Sequin formatted files. tRNA genes were further predicted by tRNAscan (http://lowelab.ucsc.edu/tRNAscan-SE/) (Supplementary Table 43).

**Transcriptome library and gene expression analysis**. The paired-end reads for protocorm, juvenile tuber, immature tuber, mature tuber, scape samples were mapped to the $G.$ $elata$ genome using TopHat. The total numbers of aligned reads were normalized by gene length and sequencing depth for an accurate estimation of expression level. We used these normalized read counts (RPKM) as the expression level for each gene. Then, DESeq was used to identify differentially expressed genes (Supplementary Tables 44-58). A repeated bisection method and a top–down hierarchical clustering algorithm in gCLUTO (http://glaros.dtc.umn.edu/gkhome/cluto/gcluto/overview) were used to generate the expression profiles of all differently expressed genes.

**RNA extraction and quantitative PCR**. Four different $G.$ $elata$ tissues (epidermis, cortex, parenchymal cell A and B) were collected from three mature tuber samples. Total RNAs were extracted using TRIzol® Reagent (Thermo Fisher Scientific, USA) according to the manufacturer's instructions. The concentrations and purities of the total RNAs were assessed by a spectrophotometric analysis at 260 and 280 nm. One μg of total RNA was reverse transcribed at 42 °C using TransScript® Reverse Transcriptase (TransBionovo Co, China) and Oligo(dT)$_{18}$ according to the manufacturer's recommendations. Prior to use in qPCR, cDNA was diluted 1:5 with H$_2$O.

The qPCR reactions were performed in duplicate for each condition using the KAPA SYBR® FAST qPCR Master Mix (KapaBiosystems, USA) and LightCycler® 480 Real-Time PCR System (Roche, Switzerland). Each reaction consisted of 20 μL containing 1 μL of cDNA and 200 nM of each primer (Supplementary Tables 59 and 60). The cycling conditions were: denaturation at 95 °C for 3 min; followed by 45 two-segment cycles of amplification at 95 °C for 10 s, and 60 °C for 30 s in which fluorescence was automatically measured, and one three-segment cycle of 95 °C for 5 s, 65 °C for 1 min, and 95 °C for 30 s. The baseline adjustment method of the LightCycler® 480 software was used to determine the Ct in each reaction. β-actin was selected as the internal control and the expression levels of tested genes were determined using the comparative Ct ($2^{-\Delta\Delta Ct}$) method.

**Diversity of microbial communities**. Four different $G.$ $elata$ tissues (protocorm, juvenile tuber, immature tuber, mature tuber) were collected and total genomic DNAs were extracted using hexadecyl trimethyl ammonium bromide (CTAB). The 16S V4 and ITS1 genes in all sample were amplified using the universal primers 515F-806R and ITS5-1737F with a barcode as a marker for distinguishing samples. The PCR was performed with Phusion® High-Fidelity PCR Master Mix (New England Biolabs, Ipswich, MA, UK). PCR products were mixed in equidensity ratios. Then, the mixed PCR products were purified with Qiagen Gel Extraction Kit (Qiagen, Germany). Sequencing libraries were generated using a TruSeq® DNA PCR-Free Sample Preparation Kit (Illumina, USA) following the manufacturer's recommendations and index codes were added. The library quality was assessed on a Qubit@ 2.0 Fluorometer (Thermo Scientific) and Agilent Bioanalyzer 2100 system. The library was sequenced on an Illumina HiSeq 2500 platform and 250 bp paired-end reads were generated.

High-quality sequences were clustered into OTUs defined at 97% similarity. These OTUs were applied for diversity, richness and rarefaction curve analyses using MOTHUR. Taxonomic assignments of OTUs that reached the 97% similarity level were made using the QIIME (quantitative insights into microbial ecology) software package through comparison with the SILVA, Greengene, and RDP databases. Venn diagrams were generated to identify the mutual and specific taxons between groups using R software (http://www.r-project.org/).

**Microscopy**. For identification and analysis of $G.$ $elata$ infected by $A.$ $mellea$, hand sections were cut through the infection point of an immature tuber, and the sections were then embedded in agar plates. Images were captured using a Zeiss AX10 fluorescence microscope with ×10 water immersion lenses. Owing to the

spontaneous blue fluorescence, both visible and DAPI filters were used to observe the hyphae of $A.$ $mellea$ with fluorescent microscopy (Zeiss AX10).

To analyze tissue and cell structures of $G.$ $elata$ uninfected by $A.$ $mellea$, paraffin sections (10 μm thickness) were obtained using a Thermo Scientific MicRoM HM 325 sliding microtome. For light microscopic observations of the highly lignified hyphae of $A.$ $mellea$, sections were stained with Fast Green stain reagent to investigate the infected cells of $G.$ $elata$. After staining, sections were washed by PBS three times, dehydrated through an alcohol (50%, 80%, 90%, 95%, 100%), cleared in xylene, sealed with neutral gum, and observed using a fluorescence microscope (Zeiss AX10).

**Quantification of $p$-HA and $S$-($p$-HA)-glutathione**. About 0.35 g of frozen fresh $G.$ $elata$ were homogenized and ultrasonically extracted for 30 min in four volumes (g mL$^{-1}$) water. After centrifugation (13,000 rpm, 10 min), 100 μL aliquots of supernatant were mixed with 10 μl of rutin (101.0 μg mL$^{-1}$) for quantification of $p$-HA ($\lambda$ 270 nm, Rt 6.8 min) and $S$-($p$-HA)-glutathione ($m/z$ 412.12, Rt 8.9 min) using an UPLC-PDA-ESI-Q-TOF-MSE method[73]. The injection volume was 1 μL. The contents were determined by the peak intensity ratios of the analyte to rutin ($m/z$ 609.14, 15.5 min).

**Hyphal-branching assay**. Hyphal branching in $A.$ $mellea$ fungi was evaluated in vitro by the paper disk diffusion method[74]. Primary hyphae were cultured in PDA medium containing 20 g L$^{-1}$ glucose, 4 g L$^{-1}$ potato powder and 14 g L$^{-1}$ agar. The dishes were cultured in the dark for 5–7 days at 23 °C. Secondary hyphae emerge from primary hyphae and grow upward in a negative geotropic manner in the gel; the growth of secondary hyphae was used for the assay. Test samples were first dissolved in acetone then diluted with 70% ethanol in water. The concentration of test sample solutions of natural 5-deoxy-strigol was adjusted with reference to the calibration of synthetic (±)-5-deoxy-strigol in an HPLC analysis. Paper disks (1 cm in length, 8 mm in width) loaded with 15 μL of test sample solution were placed in front of the tips of the secondary hyphae. The control was on the opposite direction of the paper without 5-deoxy-strigol. Hyphal branch patterns were analyzed at 24 h and 48 h after treatment. The sample was scored as positive for hyphal branching if new hyphal branches formed from the treated secondary hyphae. The assay was repeated at least twice, using between three and five dishes for each concentration.

**Data availability**. Genome data were deposited in GenBank under accession number PVEL00000000 and transcriptome sequence reads were deposited in the Sequence Read Archive (SRA) under accession number SRX2879747. The standard flowgram format (SFF) files related with bacterial and fungal communities were also deposited in the SRA under study accession SRX2875242 and SRX2876148. Plastid genome data were deposited in GenBank under accession number MF163256. Mitochondrial genome data were deposited in GenBank under accession numbers MF070084-MF070102.

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

## Acknowledgements

This research was funded by the Fundamental Research Funds for the Central public welfare research institutes (ZZ10-008) and Key project at central government level for the ability establishment of sustainable use for valuable Chinese medicine resources (2060302).

## Author contributions

L.H. is the lead investigator. Y.Y. and W.J. coordinated the project and directed work on the reference genome of *G. elata*. X.J., D.W., and X.M. assembled the plastid and mitochondrial genomes and analyzed gene duplication. X.Z., W.X., M.Z., Z.J., R.L. assembled the reference genome and analyzed genome characterization. J.L. performed experiments of diversity of microbial communities in *G. elata* during different periods. Y.Y., J.L., X.Z., and L.Z. performed bioinformatic analyzes to annotate the reference genome. J.Z. and C.J. extracted high-quality RNA and qPCR analysis. C.L., T.N., Z.Z., and J.Y. conceived and conducted the analyses of the metabolomics. X.W. and H.P. observed the microscopic characterization of *G. elata*. J.H., Y.Y., and Y.L. worked on culture of *Armillaria mellea* and the hyphal-branching assay. D.L., L.W., Y.Y., and Y.Z. collected samples. Y.J. and H.M. verified the gene loss using PCR.

## Additional information

**Competing interests:** The authors declare no competing interests.

