## [Peer Review File · Nature Communications]

Reviewers' comments:

Reviewer #1 (Remarks to the Author):

This is an impressive body of work, and as far as I know the first sequenced genome of a fully heterotrophic plant. As such it is a landmark paper that will influence future work on such plants, and it gives important insights into the major changes that can happen during this major shift in life history.

As with most such studies, there is also an impressive and extended set of supplementary and extended figures and tables for the reader to wade through -- a lot!! These were sometimes not well explained (acronyms, technical language, etc), and so I have made fairly substantial corrections and suggestions in the main text and this other material, to try to improve the experience for the average reader...

I have also included a fairly large number of comments and questions that should be addressed in revision, in addition to several minor and major concerns outlined below:

Major concerns --

(1) Two other orchids were used as reference genomes. *Ananas comosus*, which you used extensively in your comparisons, is also mycorrhizal (Wang and Qiu 2006, *Mycorrhiza* 16:299-363). Because mycorrhizal plants need to do some of the same things that mycoheterotrophs do (e.g., attracting fungi), it would have been useful to have a non-mycorrhizal plant as a sort of control... Anyway, I think you could think through this problem a bit in the text.

(2) I was not entirely convinced by the significance of some of the gene expansion numbers (e.g., 3 vs. 1 copy of *ArgJ*; *ANT* nos. compared similar to other orchids). It's important not to drift into 'just so' stories just because some gene numbers are different. Actually, what would it take to measure the significance of these differences? Perhaps a much broader taxon sampling? How would you test that? I think you could discuss this problem in the text, even if you can't solve it.

Additional minor concerns (**please also respond to those noted using 'Track Changes'/comments in Word)--

-- l. 64-76. This is a very technical explanation that doesn't really explain the biological basis of the pseudogene categories. Please fix this!

-- l. 67. Please submit your perl scripts to github or a similar public venue

- l. 88-92. You should say something like '... a supermatrix of concatenated genes was analyzed using RAxML.' Did you do the ML analysis on DNA or AA data (protein sequences could mean either). State what substitution models were used -- and how model testing was done to decide on the appropriate AA or DNA substitution models. Please include bootstrap or other branch support values on the tree in Fig. 1. For MCMCTREE, state the name of the calibration fossil and its associated calibration date, and cite the fossil calibration reference. Please do not say 'to get a more accurate result' as it is actually not clear which method is better

-- l. 105-155. Couldn't you just have pulled these organelles out of your main genome data?

Other fixes needed in the Methods section:

l. 18. 'Paired end'

l. 19. 'In total'

- I. 21. Delete 'The tool of'
- I. 33. 'less than'??
- I. 34. 'belong'
- I. 34. Instead of 'fungus symbiotic' just name the species
- I. 40. 'were obtained' ... 'were identified'
- I. 41. 'the other' ... 'their transcriptomes'
- I. 56. 'to make'??
- I. 61. 'filtered out'??
- I. 75. 'less than' ... add name of species before 'reference genome'
- I. 86. Delete 'was'
- I. 94. 'to have undergone'
- I. 99-100. Explain pol, gag, env
- I. 102. 'thrown away that did not contain protein domains'
- I. 103. 'with less'
- I. 152. Delete 'besides'
- I. 167. 'cells' in both cases, not 'cell'
- I. 169 & 173. Apostrophe or not for 'manufacturers'? Be consistent
- I. 174. 'the cDNA'
- I. 189. Cite Doyle and Doyle 1987 classic paper for CTAB method
- I. 207. Which R script?
- I. 211. Delete 'the'

Please find other comments from this reviewer in the 4 files attached.

Reviewer #2 (Remarks to the Author):

This paper reports the genome sequence of the mycoheterotrophic orchid *Gastrodia*. The principal interest of this paper, in my opinion, is the determination that the gene space in the species is possibly much smaller than in other angiosperms - only about 19,000 annotated genes. As such, the argument is largely loss of many genes present in common ancestors with phototrophic orchids. However, demonstration of gene loss (as opposed to gene gain) requires extraordinary effort. The *Zostera* (sea grass) paper in *Nature* dealt with similar issues of gene loss; the present work should be held to no less stringency. The present authors report having sequenced a number of *Gastrodia* transcriptomes and using other species' gene models for prediction training purposes. They discuss statistics from BUSCO regarding completeness of the annotation, and report evidence that photosynthetic BUSCOs are those enriched among genes not detected - Extended Data Table 12. Only two GO terms are reported, both of which are cellular component terms indirectly related to photosynthesis. I cannot find description of the statistics performed here. What was the statistical background used? (Indeed, all statistical tests need to be completely described.) What correction for multiple tests was used? Are these two terms really the only 2 GOs that are enriched among lost BUSCOs? Have the authors tried enrichment analysis using only biological process GOs? Use of cellular component GO terms is extremely gross and much less convincing than would be biological process GOs. The authors' case for reduction of photosynthetic BUSCOs, in my opinion, depends on proper statistical determination of photosynthetic biological process enrichment among lost genes. The BUSCO set is an arbitrary set, not a biological one, so it will not otherwise be correct to cherry-pick missing BUSCOs related to photosynthesis; for a true statistical test to work, the effect must be pronounced more than random chance would predict, using the "randomly functioning" BUSCO set as a random sample of purportedly conserved genes among angiosperms. Furthermore, the authors do not describe additional attempts to locate particular genes on their assembly that might not be represented by complete gene models of called pseudogenes. In my opinion, it is absolutely necessary

that the authors search through the naked assembly for sequences with high sequence similarity to genes they conclude to be missing. They may discover partials in this way that could actually be interesting as regards the process of gene loss, or they could (perhaps likely for some) discover sequences that upon manual gene model reproduction provide full-length genes. The *Gastrodia* assembly is highly contiguous at the scaffold level, but less so at the contig level (N50~69Kb), implying lots of sequencing gaps - of course, such gaps could lead to a technical (and not biological) decrease in the number of genes called. While I do "believe" that a mycoheterotrophic plant such as *Gastrodia* could likely lose some photosynthetic genes that were truly photosynthesis-unique, the burden of proof for a high profile paper is extreme. I would strongly suggest that the authors design PCR primers for highly conserved regions of a selected set (maybe 10-20) of the most biologically interesting "missing genes" - these could be evaluated for their likelihood of amplification success by comparison with existing orchid and other monocot genomes. Then try PCR on *Gastrodia* and other orchid species' nuclear DNA for positive controls. If the authors actually amplify any of the 10-20 "missing genes" from *Gastrodia*, then perhaps their informatically based conclusions on absence could be questioned. But if not, this might become supportive, but of course still negative evidence only. The work on expansions and contractions of gene families does not attempt a statistical perspective, so far as I can see. Other recent papers have successfully used the application BadiRate for this purpose. For example, the difference between 1 vs. 3 family members might be insignificant compared to 3 vs. 12 family members. Note particularly that gene "families" such as determined here are mostly artificial sequence similarity clusters as opposed to true phylogenetic groups - often, sequence-divergent singletons (either biologically real, or poor gene model calls) are cast out of such orthogroups by the algorithm despite actually belonging in a true family. As such, the authors must search through singletons as well for possible gene family members of highlighted families, and perform phylogenetic analyses that might clearly show loss of specific parlous. The metagenomic analyses for associated microbiome in the paper seem misplaced, but regardless, it is not apparent that they are up to technical standard with multiple biological and technical replicates. The plastid genome is reduced, as expected based on many other papers on mycoheterotrophic plastid genomes; I do not understand the connection that the authors draw here: "It is reasonable to speculate that these expanded genes likely relate in some way to the functional requirements of the unique obligate myco-heterotrophic lifestyle of *G. elata*, and we subsequently sequence and assemble the mitochondrial genome of *G. elata* to explore this idea." This does not follow for me.

Reviewer #3 (Remarks to the Author):

These paper reports on the genome of *Gastrodia elata*, an orchid that depends on a fungus to get N, P and C resources. Although this plant has a relatively large genome (1 Gb), it apparently has the smallest proteome of all angiosperms sequenced to data. The work presented is quite interesting and presents evidence that genome plasticity explains, at least partially, the evolutive adaptations to a heterotrophic form life that depends on symbiosis with a fungus. However, there are several parts of the description of the genome that need to be worked out and explained in more detail for this paper to be acceptable for publication. I have a number of concerns that the authors should address to complete the description of the *G elata* genome:

1- There is important methodological information missing. For instance, I cannot find which sequencing platform was used to sequence the *G. elate* genome. Also, it is not clear whether the analysis of the analysis of Single-Copy Orthologs analysis was done using the assembled transcriptome or the annotated gene in the genome

2- It is also difficult to understand how the BUSCO gene set analysis tool was used to assess the

quality of the genome given that many conserved genes are missing. It is important to explain more clearly gene prediction were made and how the quality of the genome sequence and assembly was estimated.

3- It will be interesting to include more information about the five groups of differentially expressed genes that were identified for each of the *G. elata* developmental stages. Do they represent different rates of metabolic activity, cell division or cell differentiation?

4- The authors show that 67% of the *G. elata* genome is represented by repetitive elements, that corresponds to about 670 Mb out of the 1.06 Gb genome size. The coding part of the genome, considering a gene size of 3 kb in average, correspond to an additional 57 Mb. What are the remain 275 Mb of the *G. elata* genome? Does this part of the genome comprises non-coding RNAs, pseudogenes, long introns, long intergenic regions? It would be interesting to have a more detailed description of the genome.

5- The authors state that many highly conserved genes in other species have been deleted in *G. elata*. To make such conclusion, it is important to have a more detailed syntenic analysis with other closely related species to have a better understanding of the mechanisms that lead to the loss of genes. Have these genes been deleted or is the genome full of pseudogenes? If genes have been deleted, how these deletions happened? Are there solo-LTR flanking the deleted regions?

6- The papers need to be completely revised by a native English-speaking person to correct grammar and spelling mistakes

Dear Editor,

On behalf of my co-authors, we thank you very much for giving us an opportunity to revise our manuscript, we appreciate editor and reviewers very much for their positive and constructive comments and suggestions on our manuscript entitled “The *Gastrodia elata* genome provides insights into orchid adaptation to heterotrophy” (ID: NCOMMS-17-15288-T).

We have studied reviewer’s comments carefully and have made revision which included a point by point response to the reviewers’ comments as following, and revised manuscript using 'Track Changes'/comments in Word. We have tried our best to revise our manuscript according to the comments. Attached please find the revised version, which we would like to submit for your kind consideration.

Reviewer 1:

This is an impressive body of work, and as far as I know the first sequenced genome of a fully heterotrophic plant. As such it is a landmark paper that will influence future work on such plants, and it gives important insights into the major changes that can happen during this major shift in life history.

As with most such studies, there is also an impressive and extended set of supplementary and extended figures and tables for the reader to wade through -- a lot!! These were sometimes not well explained (acronyms, technical language, etc), and so I have made fairly substantial corrections and suggestions in the main text and this other material, to try to improve the experience for the average reader...

I have also included a fairly large number of comments and questions that should be addressed in revision, in addition to several minor and major concerns outlined below:

Major concerns:

(1) Two other orchids were used as reference genomes. *Ananas comosus*, which you used extensively in your comparisons, is also mycorrhizal (Wang and Qiu 2006, *Mycorrhiza* 16:299-363). Because mycorrhizal plants need to do some of the same things that mycoheterotrophs do (e.g., attracting fungi), it would have been useful to have a non-mycorrhizal plant as a sort of control... Anyway, I think you could think through this problem a bit in the text.

Thanks for the reviewer’s good suggestion. We would like to use *Arabidopsis thaliana*, which is a non-mycorrhizal plant, as a sort of control plant. All the data about *A. thaliana* have been added in revised paper and the supplementary materials.

(2) I was not entirely convinced by the significance of some of the gene expansion numbers (e.g., 3 vs. 1 copy of ArgJ; ANT nos. compared similar to other orchids). It's important not to drift into 'just so' stories just because some gene numbers are different. Actually, what would it take to measure the significance of these differences? Perhaps a much broader taxon sampling? How would you test that?

I think you could discuss this problem in the text, even if you can't solve it.

Thanks for the reviewer's good suggestion. Firstly, we have all added the Fisher's test to show the importance of the gene expansion and contraction. Fisher's exact test show that the gene number of ArgJ in *G. elata* genome is expanded compared to *P. equestris*, *D. officinale*, *A. comosus* and *A. thaliana* ($p < 0.05$). Secondly, we have cut off those expanded genes ($p > 0.05$), such as ANT and α -glucosidases.

Additional minor concerns (**please also respond to those noted using 'Track Changes'/comments in Word):

-- l. 64-76. This is a very technical explanation that doesn't really explain the biological basis of the pseudogene categories. Please fix this!

Thank you for your suggestions, we rewritten pseudogene identification methods in the revised paper, which we gave biological explanations for pseudogene categories:

Line 415 "Pseudogenes in the *G. elata* genome were identified by searching against *G. elata* intergenic regions using *D. officinale* or *P. equestris* protein sequences as the seed sequences (TBLASTN, E-value cut off $1e^{-5}$). Before the BLAST search, regions of the 18,969 true genes were masked. The BLAST hits were conjoined by Solar software. GeneWise was used to predict the pseudogene structures with the '-pseudo' parameter. Pseudogenes were then classified by PseudoPipe. The PseudoPipe program applies a set of sequence identity and completeness cut off to report a final set of good-quality pseudogene sequences. We used the following cutoffs: amino acid (AA) sequence identity $> 30\%$ and match length > 50 AA to filter out false positives. GeneWise results that fulfilled the cut off criteria were denoted as high-confidence pseudogenes. High-confidence pseudogenes were then assigned to three categories. (1) Processed/retrotransposed pseudogenes (PSSDs), which formed through retrotransposition. Retrotransposition occurred by reintegration of a cDNA, a reverse transcribed mRNA transcript, into the genome at a new location. (2) Duplicated pseudogenes (DUPs), which formed through gene duplication, following by decay of genes, include frameshifts or premature stop codons. (3) Pseudogenic fragments (FRAGs), which were fragments that have high-sequence similarity to known proteins, but were too decayed to be reliably assessed as processed or duplicated. We used the following criteria to classify PSSDs, DUPs and FRAGs: (i) PSSDs, exon number = 1, $0.7 < \text{align ratio} \leq 0.95$, $0.3 \leq \text{identity} \leq 0.95$; (ii) DUPs, exon number > 1 , $0.3 \leq \text{identity} \leq 0.95$, and existing insertion, deletion, termination, or frameshift; (iii) FRAGs, exon number = 1, $\text{align ratio} < 0.7$, $0.3 \leq \text{identity} \leq 0.95$."

-- l. 67. Please submit your perl scripts to github or a similar public venue.

The perl script for classification of pseudogenes came from PseudoPipe, we add reference for PseudoPipe in the revised paper.

[43. Zhang, Z. *et al.* PseudoPipe: an automated pseudogene identification pipeline. *Bioinformatics* **22**, 1437-1439, doi:10.1093/bioinformatics/btl116 (2006).]

-- 1. 88-92. You should say something like '... a supermatrix of concatenated genes was analyzed using RAxML.' Did you do the ML analysis on DNA or AA data (protein sequences could mean either). State what substitution models were used -- and how model testing was done to decide on the appropriate AA or DNA substitution models. Please include bootstrap or other branch support values on the tree in Fig. 1. For MCMCTREE, state the name of the calibration fossil and its associated calibration date, and cite the fossil calibration reference. Please do not say 'to get a more accurate result' as it is actually not clear which method is better.

Thank you for your suggestions, we have rewritten phylogenetic tree construction methods, adding model selection process, and bootstrap values were added to Fig. 1b as suggested. We also rewritten methods for divergence time estimations, Description of five calibration time points were added as suggested.

The rewritten paragraph described phylogenetic tree construction and divergence time estimation:

Line 451“Protein sequences from 74 single copy gene families were used for phylogenetic tree reconstruction. MUSCLE⁴⁹ was used to generate multiple sequence alignment for protein sequences in each single-copy family with default parameters. Then, the alignments of each family were concatenated to a super alignment matrix. The super alignment matrix was used for phylogenetic tree reconstruction through Maximum likelihood (ML) methods. Before ML reconstruction, we used ProtTest to select the best substitution models. The JTT+I+G+F model was selected as the best fit model, and RAxML was used to reconstruct the phylogenetic tree^{50,51}.

Divergence time between 14 species was estimated using McMcree in PAML⁵² with the options ‘correlated molecular clock’ and ‘JC69’ model. A Markov Chain Monte Carlo analysis was run for 20,000 generations, using a burn-in of 1000 iterations. Five calibration points were applied in the present study (**Fig. 1**): *P. equestris* and *D. officinale* divergence time (47~52.9 million years ago)⁵³, *O. sativa* and *Z. mays* divergence time (24~84 million years ago)^{54,55}, *A. thaliana* and *P. trichocarpa* divergence time (65~89 million years ago)^{56,57}, *P. trichocarpa* and *G. max* divergence time (56~89 million years ago)^{55,56}, and, root of land plants (407~557 million years ago)⁵⁶.”

-- 1. 105-155. Couldn't you just have pulled these organelles out of your main genome data?

We sequenced the genomes of two organelles of fully heterotrophic *Gastrodia elata* in order to understand plant adaptation to heterotrophy. *G. elata* have evolved a special type of plant-fungi symbiosis in which a plant gets fixed carbon and other nutrients from fungal partners, rather than from photosynthesis. As *G. elata* does not perform photosynthesis, genes related to the photosynthetic apparatus lost. We subsequently sequence and assemble the plastid genome of *G. elata* to know whether there is coevolution between plastid and nuclear genome, especially the loss of genes coding for photosynthesis systems.

We also found the expanded were enriched for several GO terms related to metabolic processes, many were all related with mitochondrial function. Mitochondria play a central role in cellular energy provision, which contains its own genome with a modified genetic code. Thus, we subsequently sequence and assemble the mitochondrial genome of *G. elata* to explore the characterization of myco-heterotrophic lifestyle of *G. elata*.

Other fixes needed in the Methods section:

1. 18. 'Paired end'
1. 19. 'In total'
1. 21. Delete 'The tool of'
1. 33. 'less than'??
1. 34. 'belong'
1. 34. Instead of 'fungus symbiotic' just name the species
1. 40. 'were obtained' ... 'were identified'
1. 41. 'the other' ... 'their transcriptomes'
1. 56. 'to make'??
1. 61. 'filtered out'??
1. 75. 'less than' ... add name of species before 'reference genome'
1. 86. Delete 'was'
1. 94. 'to have undergone'
1. 99-100. Explain pol, gag, env
1. 102. 'thrown away that did not contain protein domains'
1. 103. 'with less'
1. 152. Delete 'besides'
1. 167. 'cells' in both cases, not 'cell'
1. 169 & 173. Apostrophe or not for 'manufacturers'? Be consistent
1. 174. 'the cDNA'
1. 189. Cite Doyle and Doyle 1987 classic paper for CTAB method
1. 207. Which R script?
1. 211. Delete 'the'

Thank you for kindly reminding and we have revised our paper as your reminding. And we have used 'Track Changes'/comments in Word to response those comments.

- (1) I suggest moving 'orchid' in title to be the second word, because the study has broader implications (for understanding evolution of heterotrophy in general, not just orchid heterotrophs)
According the suggestion of reviewer, we have adjusted our title as 'The *Gastrodia elata* genome provides insights into plant adaptation to heterotrophy'.

- (2) Please include page numbers – some of us still like to read print-outs, and it helps keep track of things...

Thank you for the reviewer's kind reminding of page numbers. We have added both page numbers and line numbers for more conveniently reading.

- (3) It is controversial (because it's never been tested) whether the relationship is completely parasitic (so, perhaps the plants give something in return, who knows?). So workers in the field avoid calling this parasitism.

Yes, the reviewer is right. We have changed 'parasitism of fungi' into 'fungal partners' according to the reviewer's suggestion.

- (4) Actually, *G. elata* do have reduced leaves (=bracts) – take a look at pictures of the scape --- although field guides and orchid systematists like to pretend they are leafless!

The reviewer is correct. We have change these statements as '*Gastrodia elata* (Orchidaceae) is an orchid popularly used in traditional Chinese medicine that has a fully mycoheterotrophic lifestyle with a 'leafless' and rootless growth habit (actually they have highly reduced leaves, bracts in scape, although field guides and orchid systematists like to pretend they are leafless).'

- (5) 'We also found that the plastid genome of *G. elata* was dramatically reduced in size compared to most plant species while its mitochondrial genome is one of the largest observed to date.' ---It's reduced compared to green plants but not unusual for heterotrophs...(comparable to other full mycoheterotophs).

Yes, the reviewer has more precise thinking. According to the reviewer's suggestion, we adjusted this sentence as 'We also found that the plastid genome of *G. elata* was dramatically smaller than that of green plant species while its mitochondrial genome was one of the largest observed to date' in abstract.

- (6) You should order the number of all your figs and tables so that they are sequentially numbered by reference to first use in the main text.

According to the reviewer's suggestion, we have ordered the number of all the figs and tables.

- (7) General rule: Don't start sentences with abbreviations! Spell out any of those abbreviations (NBS, PRR, PRs) on first use in text.

According to the general rule, we have spelled out those abbreviations in the beginning of the sentences and on first use in text.

NBS, the nucleotide-binding site gene family; PRR, pattern recognition receptor genes; PRs, the pathogenesis-related gene family; others.

- (8) 'Of the 35 nuclear genes coding for photosynthetic apparatus proteins (NEP), only 12 were present in the *G. elata* genome ...' . I think you can safely assume these are non-functional because they full complement of subunits is not present. You ought to comment on this here. By the way, why do some of the photosynthetic plants have such different subunit numbers for genes like ATP synthase – do these represent some subunits having duplicated genes?

The reviewer is correct. The question about "why do some of the photosynthetic plants have such different subunit numbers for genes like ATP synthase – do these represent some subunits having duplicated genes?" needs more works on photosynthetic plants, it is besides the point of this manuscript. Thus, we added this saying 'We assume that these genes were non-functional because their full complements of subunits were not present.'

- (9) As far as I can tell from your figure, its plastid genome has also lost its major Inverted Repeat (IR) region, which would account for a large fraction of the size reduction. I think you should comment on this. '... , the *G. elata* plastid genome encodes only 19 protein-coding genes (Fig. 1e)'. You should discuss where this species fits in the 'degradation ratchet' for plastid genomes of heterotrophs (pretty far along!!), and cite Barrett and Davis (2012) (2012 <http://www.amjbot.org/content/99/9/1513.full>) and Graham et al. (2016 <http://onlinelibrary.wiley.com/doi/10.1111/nph.14398/abstract>) for this concept.

The reviewer is correct. We have added these contents and the above two citations in this paragraph (Line 183) as following 'The plastid genomes of these two species comprise two single copy regions (a large and a small single-copy region) and the two identical large inverted repeats (IRs) encode 75 and 76 genes, respectively, that are most associated with photosynthesis. The *G. elata* plastid genome has lost one IR and encodes only 19 protein-coding genes (Fig. 1e), suggesting that *G. elata* is an ancient mycoheterotroph and that its plastid genome is in the last stage of a 'degradation ratchet', i.e., retention and loss of the five core nonbioenergetic genes^{15,16}.'

- (10) Explain more clearly how the Extended Data Table 16 shows '...both the plastid and nuclear genomes of *G. elata* have lost most of the genes required for photosynthesis'. I see lots of expression!

From what I can see in your figure, and as predicted by the two refs above (Barrett and Davis 2012, Graham et al. 2016) *G. elata* has retained trnE, accD, clpP, ycf1 and 2. Some or all of these non-photosynthetic/non-genetic-apparatus genes are retained in many full

heterotrophs. I think you should comment on this, because it explains the persistence of the plastid genome in this plant!

The reviewer is correct. We have added this content to explain more clearly of Supplementary Table 30, 31.

Line 192 “..., five core nonbioenergetic genes, *trnE*, *aacD*, *clpP*, *ycf1*, and *ycf2*, were moderately to highly expressed in all five stages in *G. elata* (Supplementary Table 30, 31). These results clearly show that both the plastid and nuclear genomes of *G. elata* have lost most of the genes required for photosynthesis, although the highly degraded plastome is still essential for this full mycoheterotroph.”

- (11) Explain briefly here – how should we read this Figure 2a? What are the units, if any on the axes, and what exactly is ‘semantic space’? The legend only says ‘REViGO semantic similarity scatter plot of Biology Process Gene Ontology terms’ which is not very informative for readers unfamiliar with this method, and it doesn’t explain the axes or how to interpret the figure.

Thank you for your suggestions. We add explanation of the REViGO semantic similarity plot.

Figure legends for Figure 2a were revised as follows:

“Fig 2a. REViGO semantic similarity scatter plot of Biology Process Gene Ontology terms for expanded genes in *G. elata*. In semantic spaces, the proximity between circles represents relatedness (similarity) of the GO terms. Similar GO terms are close together in the plot. The axes in the plot have no intrinsic meaning, but were used to measure pairwise similarities between GO terms. Colour indicates degree of enrichment for each process presented as the p-value from the hyper-geometric test.”

- (12) ‘The mitochondrial genome *G. elata* is dramatically expanded (1,339 kb, Fig. 2b) compared to most of the mitochondrial genomes of other sequenced seed plants¹⁵: 37 protein coding genes were annotated (Fig. 2b), and 36 of these genes had detectable expression in mature tubers (epidermis, cortex and parenchymal cell) using a tissue-specific qPCR-based analysis.’ I think this is about the usual number of mt genes in plants... so much of the expanded genome is likely junk. Does this really support your prediction of enrichment in this genome? I am not sure you need to justify presenting an mt genome in this way. It is interesting in its own right.

We found the expanded genes were enriched for several GO terms related to metabolic processes, many were all related with mitochondrial function. Mitochondria play a central role in cellular energy provision, which contains its own genome with a modified genetic code. Thus, we subsequently sequence and assemble the mitochondrial genome of *G. elata* to explore the characterization of myco-heterotrophic lifestyle of *G. elata*. The mitochondrial genome *G. elata* has two copies of *atp4* and both are highly

expressed in cortex. ATPases may contribute to the atypical energy metabolism. This result supports our idea partly. We have reordered the words to express our idea clearly.

- (13) ‘*S*-(*p*-HA)-glutathione was detected in both *G. elata* tubers and in *A. mellea* hypha sampled from tubers (Extended Data Table 19 and Extended Data Fig. 5), which putatively suggests that *G. elata* may transport this phytoalexin to *A. mellea*.’ For what purpose? For the benefit of the fungus or its detriment? I think you could speculate here.

In our opinion, *G. elata* need *A. mellea* to transport nutrition, but should also prevent the excessive growth of *A. mellea* in the meantime. Our data firstly revealed that the phytoalexin *S*-(*p*-HA)-glutathione may be transported from *G. elata* to *A. mellea*, which strongly proved the above hypothesis. According to the reviewer’s good suggestion, we added this speculation in the text:

Line 228 ‘*S*-(*p*-HA)-glutathione was detected in both *G. elata* tubers and in *A. mellea* hyphae sampled from tubers (Supplementary Fig. 11, Supplementary Table 35), which putatively suggests that *G. elata* may transport this phytoalexin to *A. mellea* and prevent the excessive growth of *A. mellea*.’

- (14) ‘Extended Table 20 and 21’ --- The indices/metrics are just thrown up here with no attempt at explanation... please fix this so a reader can understand what the numbers in the table mean (the big picture).

Yes, the reviewer is right. We have fixed the related content of the text, and added the explanation of Supplementary Tables 36 and 37 and Fig. 2d.

In Fig. 2d, we have fixed the figure legend as ‘Venn diagrams showing the number of shared and unique fungal and bacterial Operational Taxonomic Units (OTUs) based on the ITS and 16S sequence analyses in protocorms, juvenile tubers, immature tubers, and mature tubers of *G. elata*. OTUs showed the composition and abundance of the microbe species, which were defined at 3% dissimilarity.’

In Supplementary Table 36 and Table 37, the Chao1 estimator⁷⁰ and ACE estimator^{71, 72} mainly revealed the microbial community richness; Shannon index⁷³ and Simpson index^{74, 75} mainly suggested the microbial community diversity; Goods coverage^{76, 77} showed the sequencing depth and coverage; PD whole tree⁷⁸ uncovered the phylogenetic diversity.

The related content of the text was fixed as following: ‘To investigate the effect of *A. mellea* on microbial management in *G. elata*, we performed a 16S ribosomal (rRNA) and rDNA ITS sequencing analysis and found that

the diversity of bacterial and microbial species was significantly lower during the protocorm stage than at other growth stages ($P < 0.05$), which was consistent with the pattern of gene expression of *GFP* (Fig. 2c and d, Supplementary Table 36, 37). This increased diversity of bacteria and fungi during the juvenile tuber to mature tuber periods implies that a compatible mycorrhizal fungus (*A. mellea*) can affect the structure of the microbial community associated with its host and greatly reduce the antifungal and antibacterial activities as a symbiotic association with *A. mellea* is established.’

70. Chao, A., *Non-parametric estimation of the number of classes in a population*. Scandinavian Journal of Statistics, 1984. **11**: 265-270.
71. Chao, A., et al., *Estimating the number of classes via sample coverage*. Journal of the American Statistical Association, 1992. **87**: 210-217.
72. Chao, A., et al., *Estimating the number of shared species in two communities*. Statistica Sinica, 2000.**10**: 227-246.
73. <http://www.pisces-conservation.com/sdrhelp/index.html>
74. <http://folk.uio.no/ohammer/past/diversity.html>
75. <http://www.pisces-conservation.com/sdrhelp/index.html>
76. <https://www.mothur.org/wiki/Coverage>
77. http://scikit-bio.org/docs/latest/generated/generated/skbio.diversity.alpha.goods_coverage.html#skbio.diversity.alpha.goods_coverage
78. Faith, D. P., *Conservation evaluation and phylogenetic diversity*. Biol. Conserv. (1992).

(15) ‘Without the ability to perform photosynthesis, *G. elata* depends completely on its symbiotic fungus for nutrition.’ --- I note in your ext. tables 20 and 20 that *A. mellea* is not involved in its early growth, which implies, surely, that there must be another fungal partner early on? Is it know what this is (has the necessary existence of such a partner been commented on ?) ?

A. mellea is not really involved in its early growth period (protocorm). During seed germination (protocorm), *G. elata* grows in a symbiotic relationship with a compatible mycorrhizal fungus, *Mycena* spp. The seeds of *G. elata* are tiny and do not possess an endosperm, and these seeds germinate only when adequate nutrition is obtained through the digestion of the specific fungi, *Mycena* spp. (including *M. anoectochila*, *M. dendrobii*, *M. orchidicola*, *M. osmundicola*), which invades the embryonic cells of these seeds. In this paper, we focus our attention on the relationship between *G. elata* and *A. mellea*, which is the most important fungal partner. In addition, we would like to uncover how *G. elata* could succeed to manage microbial community before and after the establishing the symbiotic association with *A. mellea*.

(16) ‘These expanded GHs include α -glucosidases and endo- β -1,4-D-xylanase, suggesting that these enzymes may have become neofunctionalized to cleave fungal glycan substrates during the digestion of hyphal walls of *A. mellea* (Fig. 3a and Extended Data Table 24).’ These seem to be a lot of variation across the genes in this table in which species has the most – and the alpha-glucosidases aren’t hugely different (?). Is this a ‘just so’ story??

Thanks for the reviewer’s good suggestion. Firstly, we have all added the Fisher’s exact test to show the importance of the gene expansion and contraction. Secondly, we have cut off those not so important expanded gene α -glucosidases ($p > 0.05$).

(17) ‘Although the total number of amino acid transporter genes is reduced in *G. elata* compared to *P. equestris*, *D. officinale*, and *A. comosus*, the number of ANT1-like aromatic and neutral amino acid transporters (ANT) genes in *G. elata* is expanded to 5, compared to only 2 in *A. comosus* (Extended Data Table 25).’ But also quite high in other orchids!

According the reviewer’s suggestion and Fisher’s exact test, we have adjusted this content in the text “Given that the ANT1-like aromatic and neutral amino acid transporters (ANT)²⁶ are known to translocate arginine (Arg), which is a key component in nitrogen translocation in arbuscular mycorrhizal fungi²⁷, it seems likely that Arg in *G. elata* is related to mycoheterotrophic symbiosis (Fig. 3a, Supplementary Table 41).”.

(18) ‘While *P. equestris*, *D. officinale*, and *A. comosus* have only one copy of glutamate N-acetyltransferase (ArgJ), an enzyme of the arginine biosynthesis pathway²⁶, *G. elata* has three copies (Extended Data Fig. 4 and Extended Data Table 25).’ Also doesn’t seem ‘hugely’ different?

Thanks for the reviewer’s good suggestion. Firstly, we have all added the Fisher’s exact test to show the importance of the gene expansion and contraction. Secondly, fisher’s exact test show that the gene number of ArgJ in *G. elata* genome is expanded compared to *P. equestris*, *D. officinale*, *A. comosus* and *A. thaliana* ($p < 0.05$) (Supplementary Table 41).

(19) ‘The extensive deletion and expansion of genes from the *G. elata* genome provides a powerful example of how a plant with a fully unique heterotrophic plant lifecycle has made use of genome plasticity to achieve extensive neo-functionalization and gene loss.’ Well, it evolved ... but is its genome more plastic than others?

The plastic of *G. elata* genome can be reveal by the global gene complement reduction, almost all functional categories of genes were reduced in *G. elata*, which was specific in all studied angiosperm genomes. This sentence was revised to:

“The extensive deletion and expansion of genes, especially the globally reduce of gene complements in almost all functional categories in *G. elata*

genome, provides a powerful example of how a plant with a fully unique heterotrophic plant lifecycle has made use of genome plasticity to achieve extensive neo-functionalization and gene loss.”

(20) ‘Our results establish an unprecedented opportunity for researchers to identification and utilization of orchid association with symbiotic fungus by focusing selection on the transcriptionally dominant genes.’ This part of the sentences seems a bit garbled and didn’t make sense to me. Perhaps just say something like ‘... an unprecedented opportunity for researchers to understand how plants that have abandoned photosynthesis continue to persist and thrive’?

According to the reviewer’s good advice, we have adjusted the above saying as ‘Our results establish an unprecedented opportunity for researchers to understand how plants that have abandoned photosynthesis continue to persist and thrive’.

(21) In Fig. 2 c, what are the units? What is *GAFP*?

GAFP is the group of genes encoding the monocot mannose-binding lectin antifungal proteins in *G. elata* which have been reported in previous work (Cox, K. D., Layne, D. R., Scorza, R. & Schnabel, G. *Gastrodia* anti-fungal protein from the orchid *Gastrodia elata* confers disease resistance to root pathogens in transgenic tobacco. *Planta* **224**, 1373-1383 (2006); Sa, Q., Wang, Y., Li, W., Zhang, L., Sun, Y. The promoter of an antifungal protein gene from *Gastrodia elata* confers tissue-specific and fungus-inducible expression patterns and responds to both salicylic acid and jasmonic acid. *Plant Cell Report* **21**, 79-84 (2003)). The units means the expression levels (>1) of different gene member of *GAFP* (Gel+7 number, gene number in *G. elata* genome) in the protocorm, juvenile tuber, immature tuber, mature tuber, and scape of *G. elata*. According to the reviewer’s suggestion, we have added the above explanation in Fig. 2c legend.

(22) In Fig. 3, this is very cool. Is the pathway in (a) new here?

In Fig. 3a, this signaling pathway is commonly existed in arbuscular mycorrhiza (AM) plant, but have not been revealed in mycorrhizal plant associated with *A. mellea* (eg: *G. elata*). Interestingly, we found the most genes involved in this pathway were expanded in *G. elata* genome. Thus, we presumed strigolactone might be a putative signal compound in *G. elata* and excitingly found it could promote branching of *A. mellea* hyphae (Fig. 3b).

Reviewer 2:

This paper reports the genome sequence of the mycoheterotrophic orchid *Gastrodia*. The principal interest of this paper, in my opinion, is the determination that the gene space in the species as possibly much smaller than in other angiosperms - only about 19,000 annotated genes. As such, the argument is largely loss of many genes present in common ancestors with phototrophic orchids. However,

demonstration of gene loss (as opposed to gene gain) requires extraordinary effort. The *Zostera* (sea grass) paper in Nature dealt with similar issues of gene loss; the present work should be held to no less stringency.

Thanks for the reviewer's good suggestion. In our revised paper, we added detail description of gene loss analysis results.

Line 96 "Comparison of the sequenced genomes of the orchid species *G. elata*, *P. equestris*⁷, and *D. officinale*^{8,9} indicated that they diverged approximately 67 million years ago (Fig. 1b and Supplementary Fig. 4). Two ancient whole genome duplication (WGD) events are evident in the *G. elata* genome; these events can also be discerned in the genomes of *P. equestris* and *D. officinale* suggesting they occurred prior to the divergence of the three orchid species (Supplementary Fig. 5). Compared to *P. equestris* (29,431 protein coding genes) and *D. officinale* (28,910 protein coding genes), *G. elata* has a relative small proteome size (18,969 protein coding genes). The estimated proteome size of *G. elata* is the smallest theoretical proteome so far identified among angiosperm genomes (Supplementary Table 12). Comparison of *G. elata*, *P. equestris* and *D. officinale* genes that have functional annotation information revealed global gene set reduction in the *G. elata* genome. For example, almost all second level Gene Ontology (GO) categories had fewer genes in *G. elata* than in the other two species, and 14 of these categories (25.9%) were significant reduced (Fisher's exact test, $p < 0.05$, Supplementary Fig. 6 and Supplementary Table 13). We also found that several Pfam domain families were significantly reduced in the *G. elata* genome (Supplementary Table 14). Among the 14 angiosperm used in the phylogenetic analysis, *G. elata* had the lowest number of gene families; moreover, *G. elata* had on average the lowest number of genes in each gene family (Fig. 1c, and Supplementary Table 15). This consistently low number of genes and gene families suggests that many gene families have been eliminated from the *G. elata* genome, and further suggests that many of the remaining gene families have contracted. Gene family expansion and contraction analysis based on maximum likelihood modeling of gene gain and loss confirmed that many more gene families had undergone contraction in *G. elata* compared to the other two orchid genomes (Supplementary Fig. 7 and Supplementary Table 16). A Benchmarking Universal Single-Copy Orthologs (BUSCO) analysis, which assessed 956 orthologous groups with genes present as single-copy in at least 90% of plant genomes¹⁰, revealed that 195 (20.4%) highly-conserved genes were missing from the *G. elata* genome. This rate of absence is much higher than in the genomes of the 14 land species that were included in this analysis (Supplementary Table 17). All of these analyses indicate that *G. elata* has undergone extensive gene losses, even for genes that were conserved in other plant species that have also undergone extensive lost events.

The absence of these genes is unlikely to be due to genome assembly problems because 98.66% of the transcripts assembled from transcriptome data could be mapped to the assembly. Another possibility is that several genes were missed due to gene prediction problems. By mapping RNA reads onto the annotated genome, we found that the majority of RNA reads (>86%) from all *G. elata* tissues could be mapped to annotated exon regions (Supplementary Table 18). This rate of mapping

was comparable to that achieved in the well-annotated rice genome and higher than in the *P. equestris* genome (Supplementary Table 18). Through analysis of gene synteny among *G. elata* and *P. equestris* and *D. officinale*, we detected 2961 gene deletion events in *G. elata* versus *P. equestris*, and 3120 gene deletion events in *G. elata* versus *D. officinale* (Supplementary Table 19). Further TBLASTN searches of these deleted genes recovered less than 3% of them. Of these genes, fewer than 15% were supported by RNA-seq data (Supplementary Table 19). Both the RNA mapping results and the synteny deletion analysis confirmed that our gene prediction was comprehensive; thus, the possibility of missing gene annotations was low. Finally, PCR amplification of 18 lost genes (*atpD*, *atpG*, *IhcA*, *IhcB*, *psaD*, *psaF*, *psaL*, *psaN*, *psbO*, *psbR*, *psbY*, *psb27*, *psb28*, *petC*, *petE*, *ICS*, *DHAR*, and *TRX*) confirmed that all were absent from the *G. elata* genome (Supplementary Fig. 8 and Supplementary Table 20). Thus, the global gene losses in *G. elata* represent evolutionary events, and might be the result of adaption to an obligate mycoheterotrophic lifestyle.”

The present authors report having sequenced a number of *Gastrodia* transcriptomes and using other species’ gene models for prediction training purposes. They discuss statistics from BUSCO regarding completeness of the annotation, and report evidence that photosynthetic BUSCOs are those enriched among genes not detected - Extended Data Table 12. Only two GO terms are reported, both of which are cellular component terms indirectly related to photosynthesis. I cannot find description of the statistics performed here. What was the statistical background used? (Indeed, all statistical tests need to be completely described.) What correction for multiple tests was used? Are these two terms really the only 2 GOs that are enriched among lost BUSCOs? Have the authors tried enrichment analysis using only biological process GOs? Use of cellular component GO terms is extremely gross and much less convincing than would be biological process GOs. The authors’ case for reduction of photosynthetic BUSCOs, in my opinion, depends on proper statistical determination of photosynthetic biological process enrichment among lost genes. The BUSCO set is an arbitrary set, not a biological one, so it will not otherwise be correct to cherry-pick missing BUSCOs related to photosynthesis; for a true statistical test to work, the effect must be pronounced more than random chance would predict, using the “randomly functioning” BUSCO set as a random sample of purportedly conserved genes among angiosperms.

We are sorry that the written here was confusing. We use BUSCO to describe GO enrichment because we try to say that conserved genes are also lost in *G. elata* genome. To resolve this confusing description, we rewrote paragraphs about gene lost. Under-represent of genes in *G. elata* was now tested using all second level GO terms.

Line 105 “Comparison of *G. elata*, *P. equestris* and *D. officinale* genes that have functional annotation information revealed global gene set reduction in the *G. elata* genome. For example, almost all second level Gene Ontology (GO) categories had fewer genes in *G. elata* than in the other two species, and 14 of these categories (25.9%) were significant reduced (Fisher’s Exact test, $p < 0.05$, Supplementary Fig. 6 and Supplementary Table 13). We also found that several Pfam domain families were

significantly reduced in the *G. elata* genome (Supplementary Table 14). Among the 14 angiosperm used in the phylogenetic analysis, *G. elata* had the lowest number of gene families; moreover, *G. elata* had on average the lowest number of genes in each gene family (Fig. 1c, and Supplementary Table 15). This consistently low number of genes and gene families suggests that many gene families have been eliminated from the *G. elata* genome, and further suggests that many of the remaining gene families have contracted. Gene family expansion and contraction analysis based on maximum likelihood modeling of gene gain and loss confirmed that 3,586 gene families had undergone contraction in *G. elata*, much more compared to the other two orchid genomes (Supplementary Fig. 7 and Supplementary Table 16). A Benchmarking Universal Single-Copy Orthologs (BUSCO) analysis, which assessed 956 orthologous groups with genes present as single-copy in at least 90% of plant genomes¹⁰, revealed that 195 (20.4%) highly-conserved genes were missing from the *G. elata* genome. This rate of absence is much higher than in the genomes of the 14 land species that were included in this analysis (Supplementary Table 17). All of these analyses indicate that *G. elata* has undergone extensive gene losses, even for genes that were conserved in other plant species that have also undergone extensive lost events.”

Furthermore, the authors do not describe additional attempts to locate particular genes on their assembly that might not be represented by complete gene models of called pseudogenes. In my opinion, it is absolutely necessary that the authors search through the naked assembly for sequences with high sequence similarity to genes they conclude to be missing. They may discover partials in this way that could actually be interesting as regards the process of gene loss, or they could (perhaps likely for some) discover sequences that upon manual gene model reproduction provide full-length genes.

We do have efforts to exclude the possibility of missing gene annotation, we added this in the method section of the revised paper.

Line 394 “Gene models only supported by ab initio evidence were filtered out. To reduce the possibility of missing and poorly annotated genes, we invested additional effort in annotating some gene families that could be missed by automated genome annotation, such as NBS-encoding genes. In total, 1943 protein sequences containing an NB-ARC domain were searched against the *G. elata* genome using TBLASN with a threshold of $1e^{-5}$. All BLAST hits in the genome, together with 5000 bp flanking regions on both sides, were annotated by the GeneWise program. The resulting predictions were surveyed to verify whether they encoded NBS or LRR motifs using Pfam. We also focused on other genes, such as those related to photosynthesis, and transporter, and these were manually annotated through a combination of BLAST search and motif verification. Ultimately, a comprehensive non-redundant reference gene set was produced that contained 18,969 protein coding gene models.”

And we use two methods to check the completeness of our annotations, which we described in the main text of the revised paper.

Line 129 “The absence of these genes is unlikely to be due to genome assembly

problems because 98.66% of the transcripts assembled from transcriptome data could be mapped to the assembly. Another possibility is that several genes were missed due to gene prediction problems. By mapping RNA reads onto the annotated genome, we found that the majority of RNA reads (>86%) from all *G. elata* tissues could be mapped to annotated exon regions (Supplementary Table 18). This rate of mapping was comparable to that achieved in the well-annotated rice genome and higher than in the *P. equestris* genome (Supplementary Table 18). Through analysis of gene synteny among *G. elata* and *P. equestris* and *D. officinale*, we detected 2961 gene deletion events in *G. elata* versus *P. equestris*, and 3120 gene deletion events in *G. elata* versus *D. officinale* (Supplementary Table 19). Further TBLASTN searches of these deleted genes recovered less than 3% of them. Of these genes, fewer than 15% were supported by RNA-seq data (Supplementary Table 19). Both the RNA mapping results and the synteny deletion analysis confirmed that our gene prediction was comprehensive; thus, the possibility of missing gene annotations was low. Finally, PCR amplification of 18 lost genes (*atpD*, *atpG*, *IhcA*, *IhcB*, *psaD*, *psaF*, *psaL*, *psaN*, *psbO*, *psbR*, *psbY*, *psb27*, *psb28*, *petC*, *petE*, *ICS*, *DHAR*, and *TRX*) confirmed that all were absent from the *G. elata* genome (Supplementary Fig. 8 and Supplementary Table 20). Thus, the global gene losses in *G. elata* represent evolutionary events, and might be the result of adaption to an obligate mycoheterotrophic lifestyle.”

The *Gastrodia* assembly is highly contiguous at the scaffold level, but less so at the contig level (N50~69Kb), implying lots of sequencing gaps - of course, such gaps could lead to a technical (and not biological) decrease in the number of genes called.

We could not rule out the possibility that in some rare cases, genes might be missed in our assembly due to technical problems like sequencing bias of Illumina Platform. This problem indeed can be put onto almost all genomes that assembled through whole genome shotgun methods.

Although the Contig N50 level of our assembly are not very high, our value is better than many published plant genomes which assembled by Illumina data. And are comparable to or even better than newly published Orchid genomes assembled by Illumina and PacBio data (Table R1).

The assembly quality of our genome were also evaluated by mapping DNA sequencing reads onto the genome, and by mapping transcripts assembled from RNA-seq data to the genome. All results suggested that our assembly quality was good.

The description of assembly quality in the revised paper:

Line 67 “98.51% of the raw sequence reads could be mapped to the assembly, suggesting that our assembly results contained comprehensive genomic information (Supplementary Table 3). Gene region completeness was evaluated by RNA-Seq data (Supplementary Table 4): of the 80,646 transcripts assembled by Trinity, 98.66% could be mapped to our genome assembly, and 94.41% were considered as complete (more than 90% of the transcript could be aligned to one continuous scaffold).”

Table R1 Contig N50 for Published Plant genomes assembled by Illumina Data.
Red lines are Orchid genomes.

Species	Published Year	Journal	Contig N50 (kb)	Scaffold N50 (Mb)
Gastrodia elata			69.0	4.912
Dendrobium officinale	2016	Scientific report	33.09	0.391
Phalaenopsis equestris	2015	Nature Genetics	20.56	0.359
Dendrobium officinale (add PacBio data)	2017	Molecular plant	51.74	1.055
Phalaenopsis equestris (add PacBio data)	2017	Nature	45.79	1.217
Apostasia shenzhenica (add PacBio data)	2017	Nature	80.07	3.029
Zostera marina	2016	Nature	79.9	0.485
Cladosiphon okamuranus	2016	DNA research	21.7	0.416
Symbiodinium microadriaticum	2016	Scientific report	34.9	0.574
Hevea brasiliensis	2016	Nature plant Nature	30.6	1.28
Manihot esculenta	2016	Biotechnology	27.5	0.648
Arachis duranensis	2016	Nature genetics	22.3	0.947
Ziziphus jujuba	2016	PLOS Genetics	34	0.754
Ginkgo biloba	2016	Gigascience	48.2	1.36
Musa itinerans	2016	Scientific report	33.9	0.192
Juglans regia	2016	The plant journal	46	0.465
Olea europaea	2016	Gigascience	52.3	0.443
Lepidium meyenii	2016	Molecular Plant	81	2.4
Siraitia grosvenorii	2016	PNAS Plant Biotechnology	34	0.101
Lupinus angustifolius	2016	Journal	45.6	0.703
Macleaya cordata	2017	Molecular Plant	25	0.308
Rhodiola crenulata	2017	Gigascience	25.4	0.145
prunus avium	2017	DNA research	N	0.219
Momordica charantia	2017	DNA research	25.1	1.1
Cicer reticulatum	2017	DNA research	23.2	0.217
Populus pruinosa	2017	Gigascience	14	0.698
Punica granatum	2017	The plant journal	67	1.89
Capsella bursapastoris	2017	The plant journal	37.9	0.628
Hibiscus syriacus	2017	DNA research	30	0.14
Rhododendron delavayi	2017	Gigascience	61.8	0.638
Camellia sinensis	2017	Molecular Plant	20	0.449
Fraxinus excelsior	2017	Nature	25	0.104

Ruellia speciosa	2017	DNA research	1.55	0.018
Ipomoea batatas	2017	Nature Plant	N	0.201
Panax notoginseng	2017	Molecular Plant Nature	16	0.096
Cenchrus americanus	2017	biotechnology	18	0.885

While I do “believe” that a mycoheterotrophic plant such as *Gastrodia* could likely lose some photosynthetic genes that were truly photosynthesis-unique, the burden of proof for a high profile paper is extreme. I would strongly suggest that the authors design PCR primers for highly conserved regions of a selected set (maybe 10-20) of the most biologically interesting “missing genes” - these could be evaluated for their likelihood of amplification success by comparison with existing orchid and other monocot genomes. Then try PCR on *Gastrodia* and other orchid species’ nuclear DNA for positive controls. If the authors actually amplify any of the 10-20 “missing genes” from *Gastrodia*, then perhaps their informatically based conclusions on absence could be questioned. But if not, this might become supportive, but of course still negative evidence only.

Thank you for your suggestions. We have design PCR primers for highly conserved regions of a selected set as following of the most biologically interesting ‘missing genes’, and then did PCR on *Gastrodia elata*, and other species (*Dendrobium officinale*, *Ananas comosus*, and *Arabidopsis thaliana*) for positive controls, and blank control as negative control. As expected, we couldn’t amplify those ‘missing genes’ from *Gastrodia elata*. Of the 18 genes we tested for, none of them could be amplified from *G. elata*, which confirmed our gene loss analysis. Those results were showed in supplementary materials (supplementary figure 8 and supplementary table 20).

The work on expansions and contractions of gene families does not attempt a statistical perspective, so far as I can see. Other recent papers have successfully used the application BadiRate for this purpose. For example, the difference between 1 vs. 3 family members might be insignificant compared to 3 vs. 12 family members.

Thank you for your suggestions. In our revised paper, CAFÉ 2.2(Computational Analysis of gene Family Evolution) was used to identify global gene family expansion and contractions (Supplementary Figure 7). In addition, for all gene families we described in the paper, we add Fisher’s exact test to examine significant gene copy number variations.

Note particularly that gene “families” such as determined here are mostly artificial sequence similarity clusters as opposed to true phylogenetic groups - often, sequence-divergent singletons (either biologically real, or poor gene model calls) are cast out of such orthogroups by the algorithm despite actually belonging in a true family. As such, the authors must search through singletons as well for possible gene family members of highlighted families, and perform phylogenetic analyses that might clearly show loss of specific parlous.

According to the reviewer's good suggestion, we have performed the phylogenetic analyses for all gene families we described in the paper. All phylogenetic trees were added in supplemental materials (Supplementary Figure 13-32).

The metagenomic analyses for associated microbiome in the paper seem misplaced, but regardless, it is not apparent that they are up to technical standard with multiple biological and technical replicates.

Yes, the reviewer is right. We have fixed the related content of the text, and added the explanation of the multiple biological and technical replicates in Extended Table 36 and 37 and Fig. 2d. The related content of the text was fixed as following: 'To investigate the effect of *A. mellea* on microbial management in *G. elata*, we performed a 16S ribosomal (rRNA) and rDNA ITS sequencing analysis and found that the diversity of bacterial and microbial species was significantly lower during the protocorm stage than at other growth stages ($P < 0.05$), which was consistent with the pattern of gene expression of *GAFP* (Fig. 2c and d, Supplementary Table 36, 37). This increased diversity of bacteria and fungi during the juvenile tuber to mature tuber periods implies that a compatible mycorrhizal fungus (*A. mellea*) can affect the structure of the microbial community associated with its host and greatly reduce the antifungal and antibacterial activities as a symbiotic association with *A. mellea* is established.'

In Fig. 2d, we have fixed the figure legend as 'Venn diagrams showing the number of shared and unique fungal and bacterial Operational Taxonomic Units (OTUs) based on the ITS and 16S sequence analyses in protocorms, juvenile tubers, immature tubers, and mature tubers of *G. elata*. OTUs showed the composition and abundance of the microbe species, which were defined at 3% dissimilarity.'

In Supplementary Table 36 and Table 37, we added the multiple biological and technical replicates in table legends as following 'Reads filtering: (1) removing reads containing more than 10% of unknown nucleotides; (2) removing reads containing less than 80% of bases with quality (Q-value) > 20 . The filtered reads were then assembled into tags according to overlap between paired-end reads with more than 10-bp overlap and less than 2% mismatch. The high-quality sequences were clustered into operational taxonomic units (OTUs) defined at 97%. Chao1 estimator⁷⁰ and ACE estimator^{71, 72} mainly revealed the microbial community richness; Shannon index⁷³ and Simpson index^{74, 75} mainly suggested the microbial community diversity; Goods coverage^{76, 77} showed the sequencing depth and coverage; PD whole tree⁷⁸ uncovered the phylogenetic diversity. Protocorm: *G. elata* lives without *A. mellea*; juvenile, immature, mature tubers: *G. elata* lives with *A. mellea*. Each value is the mean of 3 replicates (\pm SD).'

The plastid genome is reduced, as expected based on many other papers on mycoheterotrophic plastid genomes; I do not understand the connection that the authors draw here: "It is reasonable to speculate that these expanded genes likely relate in some way to the functional requirements of the unique obligate

myco-heterotrophic lifestyle of *G. elata*, and we subsequently sequence and assemble the mitochondrial genome of *G. elata* to explore this idea.” This does not follow for me.

Yes, the reviewer is right. Actually, we found the expanded genes were enriched for several GO terms related to metabolic processes, many were all related with mitochondrial function. Mitochondria play a central role in cellular energy provision, which contains its own genome with a modified genetic code. Thus, we subsequently sequence and assemble the mitochondrial genome of *G. elata* to explore the characterization of myco-heterotrophic lifestyle of *G. elata*. To make our paper read more fluently, we adjusted the above sentences as ‘We speculate that these expanded genes are related in some way to the functional requirements of the obligate mycoheterotrophic lifestyle of *G. elata*. We first sequenced and assembled the mitochondrial genome to explore this idea, and the mitochondrial genome *G. elata* is dramatically expanded in size (1339 kb, Fig. 2b) compared to the mitochondrial genomes of most other seed plants¹⁷.’

Reviewer 3:

These paper reports on the genome of *Gastrodia elata*, an orchid that depends on a fungus to get N, P and C resources. Although this plant has a relatively large genome (1 Gb), it apparently has the smallest proteome of all angiosperms sequenced to data. The work presented is quite interesting and presents evidence that genome plasticity explains, at least partially, the evolutive adaptations to a heterotrophic form life that depends on symbiosis with a fungus. However, there are several parts of the description of the genome that need to be worked out and explained in more detail for this paper to be acceptable for publication. I have a number of concerns that the authors should address to complete the description of the *G. elata* genome:

1- There is important methodological information missing. For instance, I cannot find which sequencing platform was used to sequence the *G. elate* genome. Also, it is not clear whether the analysis of the analysis of Single-Copy Orthologs analysis was done using the assembled transcriptome or the annotated gene in the genome

Thank you for your suggestions, in the revised paper, we add detail descriptions of all analysis in the method section.

2- It is also difficult to understand how the BUSCO gene set analysis tool was used to assess the quality of the genome given that many conserved genes are missing. It is important to explain more clearly gene prediction were made and how the quality of the genome sequence and assembly was estimated.

Thank you for your suggestions. Description of gene prediction in the method section was completely rewritten:

Line 367 “Gene prediction was conducted through a combination of homology-based prediction, ab initio prediction and transcriptome-based prediction methods. Protein repertoires of plants including *A. comosus*³⁷, *Amborella trichopoda*³⁸, *Arabidopsis thaliana* (phytozomev10), *Brachypodium distachyon* (phytozomev10), *D. officinale*^{8,9},

O. sativa (phytozomev10), *P. equestris*⁷, *Vitis vinifera* (phytozomev10), *Sorghum bicolor* (phytozomev10) and *Zea mays* (phytozomev10) were downloaded and mapped to the *G. elata* genome using TBLASTN (E-value $\leq 1e^{-5}$). The BLAST hits were conjoined by Solar software. GeneWise (version 2.4.1) was used to predict the exact gene structure of the corresponding genomic region on each BLAST hit. Homology predictions were denoted as “Homology-set”. RNA-seq data derived from protocorm, juvenile tuber, immature tuber, mature tuber, and scape (**Fig. 1 a**) were assembled by Trinity (version 2.0). The Trinity assembly included 183,515 contigs with an average length of 592 bp. These assembled sequences were aligned against the *G. elata* genome by PASA (Program to Assemble Spliced Alignment)³⁹. Valid transcript alignments were clustered based on genome mapping location and assembled into gene structures. Gene models created by PASA were denoted as PASA-T-set (PASA Trinity set). Besides, RNA-seq reads were directly mapped to the genome using Tophat (version 2.0.8) to identify putative exon regions and splice junctions; Cufflinks (version 2.1.1) was then used to assemble the mapped reads into gene models (Cufflinks-set). Augustus (version 2.5.5)⁴⁰, GeneID (version)⁴¹, GeneScan (version 1.0)⁴², GlimmerHMM (version 3.0.1)⁴³, and SNAP (version)⁴⁴ were also used to predict coding regions in the repeat-masked genome. Of these, Augustus, SNAP and GlimmerHMM were trained by PASA-H-set gene models. Gene models generated from all the methods were integrated by EvidenceModeler (EVM). Weights for each type of evidence were set as follows: PASA-T-set > Homology-set > Cufflinks set > Augustus > GeneID = SNAP = GlimmerHMM = GeneScan. The gene models were further updated by PASA2 to generate UTRs, alternative splicing variation information, which generated 26,872 gene models. Gene models only supported by ab initio evidence were filtered out. To reduce the possibility of missing and poorly annotated genes, we invested additional effort in annotating some gene families that could be missed by automated genome annotation, such as NBS-encoding genes. In total, 1943 protein sequences containing an NB-ARC domain were searched against the *G. elata* genome using TBLASN with a threshold of $1e^{-5}$. All BLAST hits in the genome, together with 5000 bp flanking regions on both sides, were annotated by the GeneWise program. The resulting predictions were surveyed to verify whether they encoded NBS or LRR motifs using Pfam. We also focused on other genes, such as those related to photosynthesis, and transporter, and these were manually annotated through a combination of BLAST search and motif verification. Ultimately, a comprehensive non-redundant reference gene set was produced that contained 18,969 protein coding gene models. Functional annotation of the protein coding genes was carried out using BLASTP (E-value cut off $1e^{-05}$) against two integrated protein sequencing databases, SwissProt and TrEMBL. Protein domains were annotated by searching against InterPro (Version 5.16) and Pfam

(Version 3.0) databases, using InterProScan (version 4.8) and HMMER (version 3.1b1), respectively. The Gene Ontology (GO) terms for genes were obtained from the corresponding InterPro or Pfam entry. The pathways in which the genes might be involved were assigned by BLAST against the KEGG databases (release 20150831) with the E-value cut off of 1e-05. ”

We also added description of assembly quality in the revised paper:

Line 67 “98.51% of the raw sequence reads could be mapped to the assembly, suggesting that our assembly results contained comprehensive genomic information (Supplementary Table 3). Gene region completeness was evaluated by RNA-Seq data (Supplementary Table 4): of the 80,646 transcripts assembled by Trinity, 98.66% could be mapped to our genome assembly, and 94.41% were considered as complete (more than 90% of the transcript could be aligned to one continuous scaffold). ”

Nowadays, BUSCOs are commonly used to assess assembly quality of genomes. The BUSCO assessment of *G. elata* was bad because many of them were lost in *G. elata* genome. We added paragraphs to describe this in the revised paper:

Line 121 “A Benchmarking Universal Single-Copy Orthologs (BUSCO) analysis, which assessed 956 orthologous groups with genes present as single-copy in at least 90% of plant genomes¹⁰, revealed that 195 (20.4%) highly-conserved genes were missing from the *G. elata* genome. This rate of absence is much higher than in the genomes of the 14 land species that were included in this analysis (Supplementary Table 17). All of these analyses indicate that *G. elata* has undergone extensive gene losses, even for genes that were conserved in other plant species that have also undergone extensive lost events.

The absence of these genes is unlikely to be due to genome assembly problems because 98.66% of the transcripts assembled from transcriptome data could be mapped to the assembly. Another possibility is that several genes were missed due to gene prediction problems. By mapping RNA reads onto the annotated genome, we found that the majority of RNA reads (>86%) from all *G. elata* tissues could be mapped to annotated exon regions (Supplementary Table 18). This rate of mapping was comparable to that achieved in the well-annotated rice genome and higher than in the *P. equestris* genome (Supplementary Table 18). Through analysis of gene synteny among *G. elata* and *P. equestris* and *D. officinale*, we detected 2961 gene deletion events in *G. elata* versus *P. equestris*, and 3120 gene deletion events in *G. elata* versus *D. officinale* (Supplementary Table 19). Further TBLASTN searches of these deleted genes recovered less than 3% of them. Of these genes, fewer than 15% were supported by RNA-seq data (Supplementary Table 19). Both the RNA mapping results and the synteny deletion analysis confirmed that our gene prediction was comprehensive; thus, the possibility of missing gene annotations was low. Finally, PCR amplification of 18 lost genes (*atpD*, *atpG*, *IhcA*, *IhcB*, *psaD*, *psaF*, *psaL*, *psaN*, *psbO*, *psbR*, *psbY*, *psb27*, *psb28*, *petC*, *petE*, *ICS*, *DHAR*, and *TRX*) confirmed that all were absent from the *G. elata* genome (Supplementary Fig. 8 and Supplementary Table 20). Thus, the global gene losses in *G. elata* represent evolutionary events, and might be the result of adaptation to an obligate mycoheterotrophic lifestyle.”

3- It will be interesting to include more information about the five groups of differentially expressed genes that were identified for each of the *G. elata* developmental stages. Do they represent different rates of metabolic activity, cell division or cell differentiation?

According to the reviewer suggestion, we have added the analysis of different expressed genes of the *G. elata* developmental stages.

Line 91 “Our transcriptomics analysis revealed that there were 10,548 differentially expressed genes among the five growth stages; these differentially expressed genes clustered into five distinct groups that were representative of the particular stages of growth of *G. elata* (**Supplementary Fig. 3** and **Supplementary Table 11**).”

Gene Ontology (GO) enrichment (Ashburner, Ball et al. 2000) analysis for each distinct group was illustrated In Supplementary Note. Cluster 1 and Cluster 2 genes were corresponding to the protocorm and juvenile tuber stages, where *G. elata* start to establish symbiotic relationship with fungi. Several GO categories related to chemical response were enriched in these two stages, which might participate in the symbiotic interaction with fungi partners. After the establishment of steady symbiotic relationship, *G. elata* transit to a fast underground growth stage (immature tuber stage), concordantly, Cluster 3 genes are enriched for GO terms related to active energy processing and cellular growth process. The mature tuber stage is the final underground growth stage of *G. elata*, where *G. elata* finished underground growth and prepare for over ground reproductive growth, Cluster 4 genes are enriched for “response to heat”, “vegetative to reproductive phase transition of meristem”, make sure the rapid growth of *G. elata* from July to September, and transition to reproductive stage. After emerging from underground, *G. elata* finish its reproductive process within two months, Cluster 5 genes are enriched for GO terms related to nutrition transport and cell wall growth, corresponding to the rapid over ground stem growth process during this stage. Together, these data reveal that *G. elata* consisted of highly dynamic, coordinated and clear timescale transitions of gene expression during its life cycle.

4- The authors show that 67% of the *G. elata* genome is represented by repetitive elements, that corresponds to about 670 Mb out of the 1.06 Gb genome size. The coding part of the genome, considering a gene size of 3 kb in average, correspond to an additional 57 Mb. What are the remain 275 Mb of the *G. elata* genome? Does this part of the genome comprises non-coding RNAs, pseudogenes, long introns, long intergenic regions? It would be interesting to have a more detailed description of the genome.

Thank you for your suggestions, and we added one table to describe genome compositions of *G. elata*, **Supplementary Table 24** in the revised paper.

Supplementary Table 24 | Genome composition of the orchid genomes.

Annotation	G. elata		P. equestris	
	Length (bp)	Perce nt	Length (bp)	Perce nt
Genic	293,048,29	27.62	283,578,30	26.65
	4	%	0	%
Exon	28,820,953	2.72%	26,389,941	2.48%
Intron	264,227,34	24.90	257,188,35	24.17
	1	%	9	%
Intergenic				
TEs and repeats	702,250,87	66.18	633,166,32	59.51
	6	%	0	%
LTR retrotransposon	593,527,75	55.94	491,966,30	46.24
	5	%	1	%
DNA transposon	46,521,569	4.38%	45,683,395	4.29%
Non-coding RNA	799,049	0.07%		
Pseudogene	768,453	0.07%		
N	35,579,041	3.35%	82,558,897	7.60%
Assembled genome length	1,061,091,640	---	1,064,051,384	----

5- The authors state that many highly conserved genes in other species have been deleted in *G. elata*. To make such conclusion, it is important to have a more detailed syntenic analysis with other closely related species to have a better understanding of the mechanisms that lead to the loss of genes. Have these genes been deleted or is the genome full of pseudogenes? If genes have been deleted, how these deletions happened? Are there solo-LTR flanking the deleted regions?

Thank you for your suggestions. The question here is really interesting and meaningful. However, we did not take into detail for gene lost mechanisms. Genomes between *G. elata* and published orchid genomes are relatively divergent, only a small fraction (less than 30%) of the genome could be properly aligned. (**Supplementary**

Table 25), thus we only added primitive analysis for the mechanisms of gene lost in *G. elata* in the revised paper:

Line 149 “Both pseudogenizations and genome rearrangements contributed to the gene lost process of *G. elata*. We found 876 and 1,080 pseudogenes in *G. elata* using *P. equestris* and *D. officinale* genes as seeds, respectively (**Supplementary Table 21-23**). Through a whole genome alignment between *G. elata* and *P. equestris*, we found 487 genes were lost due to local rearrangements (SV genes, **Supplementary Table 25 and 26**). Functional genes in *G. elata* were located closer to transposable elements than to pseudogenes and SV genes, suggesting that transposable element did not play significant role during the gene lost processes in *G. elata*. Thus the gene lost processes in *G. elata* might be dominated by random mutations as we found many pseudogenes in *G. elata*. ”

Supplementary Table 25 | Whole genome alignment results between *G. elata* and *P. equestris*. LastZ was used to align the genomes, with the parameter “M = 254 K = 2000 L = 3000 Y = 3000 T = 1 – gextend – nochain – gapped”.

Species	Genome size (bp)	Aligned size (bp)	Aligned percentage
G. elata	1,061,091,640	312,559,265	29.46%
P. equestris	1,064,051,384	315,236,337	29.63%

6- The papers need to be completely revised by a native English-speaking person to correct grammar and spelling mistakes

Thanks for the reviewer’s suggestion. Our reviewed paper have been completely revised by a native English-speaking person to correct grammar and spelling mistakes and we hope our reviewed work would be expressed more clearly and spontaneous.

We would like to express our great appreciation to you and reviewers for comments on our paper. Looking forward to hearing from you.

Thank you and best regards.
Yours sincerely,

Yuan Yuan

Reviewers' comments:

Reviewer #2 (Remarks to the Author):

Nearly all concerns I've had since the first round of reviews have been addressed. I am, however, concerned by the authors' presentation of statistical results, apparently without corrections applied for multiple tests. This must be addressed throughout. Some specific comments, including reminders about the multiple tests issue lie below.

(1) Re:

"Two ancient whole genome 98 duplication (WGD) events are evident in the *G. elata* genome; these events can also be discerned in the genomes of *P. equestris* and *D. officinale* suggesting they occurred prior to the divergence of the three orchid species (Supplementary Fig. 5)."

— on this point you should discuss and cite the *Apostasia* genome paper, recently out in *Nature*, which came to the same conclusion.

(2) Re:

"For example, almost all 107 second level Gene Ontology (GO) categories had fewer genes in *G. elata* than in the 108 other two species, and 14 of these categories (25.9%) were significantly reduced (Fisher's Exact test, $p < 0.05$, Supplementary Fig. 6 and Supplementary Table 13)."

— The significance threshold accepted here does not appear to have been corrected for multiple tests, or at least it is not reported as such in Suppl. Table 13. If multiple tests have not been corrected for, then the 14 categories will no doubt reduce in number, possibly few or none remaining significant.

(3) Re:

"Although the *G. elata* genome has clearly undergone extensive gene loss, we found that 430 gene families (19 by a significant margin), containing 1532 genes (184 by a significant margin), showed expansion in *G. elata* compared to *P. equestris*, *D. officinale*, and *A. comosus* (Supplementary Fig. 7 and Supplementary Table 32-33)."

— similarly, the authors must correct for multiple tests and modify conclusions accordingly

(4) The authors and editors MUST search through the ENTIRE TEXT for cases where statistical results may have been presented without correction for multiple tests, as in (2) and (3) above. It is never satisfactory to avoid this when so many GOs (for example) are being tested for enrichment all at the same time.

(5) You must not phrase the text exactly as the reviewer noted:

"(4) Actually, *G. elata* do have reduced leaves (=bracts) – take a look at pictures of the scape --- although field guides and orchid systematists like to pretend they are leafless! The reviewer is correct. We have changed these statements as '*Gastrodia elata* (Orchidaceae) is an orchid popularly used in traditional Chinese medicine that has a fully mycoheterotrophic lifestyle with a 'leafless' and rootless growth habit (actually they have highly reduced leaves, bracts in scape, although field guides and orchid systematists like to pretend they are leafless).'"

— you could say "...although field guides and systematists often refer to the plants as leafless."

Reviewers' comments:

Reviewer #2 (Remarks to the Author):

Nearly all concerns I've had since the first round of reviews have been addressed. I am, however, concerned by the authors' presentation of statistical results, apparently without corrections applied for multiple tests. This must be addressed throughout. Some specific comments, including reminders about the multiple tests issue lie below.

We thank the thoughtful suggestions from the reviewer, which obvious improved the quality of our manuscript. For the concerns of statistical results, we carefully check all supplementary tables and added multiple corrections when necessary in our revised manuscripts.

(1) Re:

“Two ancient whole genome 98 duplication (WGD) events are evident in the *G. elata* genome; these events can also 99 be discerned in the genomes of *P. equestris* and *D. officinale* suggesting they occurred 100 prior to the divergence of the three orchid species (Supplementary Fig. 5).”

— on this point you should discuss and cite the *Apostasia* genome paper, recently out in Nature, which came to the same conclusion.

Thank you for your suggestion. We cite the *Apostasia* genome paper in our revised manuscript, and the description of WGD were revised to:

Two ancient whole genome duplication (WGD) events are evident in the *G. elata* genome; these events can also be discerned in the genomes of *P. equestris* and *D. officinale* suggesting they occurred prior to the divergence of the three orchid species (Supplementary Fig. 5) The older WGD event might represent the τ WGD event¹⁰ shared by most monocots, while the younger WGD event were likely shared by all extant orchids and might contribute to the divergence of orchid, as suggested in *Apostasia shenzhenica* genome¹¹.

10 Ming, R. et al. The pineapple genome and the evolution of CAM photosynthesis. *Nature genetics* 47, 1435-1442 (2015).

11 Zhang, G. Q. et al. The *Apostasia* genome and the evolution of orchids. *Nature* 549, 379-383 (2017).

(2) Re:

“For example, almost all 107 second level Gene Ontology (GO) categories had fewer genes in *G. elata* than in the 108 other two species, and 14 of these categories (25.9%) were significant reduced 109 (Fisher's Exact test, $p < 0.05$, Supplementary Fig. 6 and

Supplementary Table 13).”

— The significance threshold accepted here does not appear to have been corrected for multiple tests, or at least it is not reported as such in Suppl. Table 13. If multiple tests have not been corrected for, then the 14 categories will no doubt reduce in number, possibly few or none remaining significant.

Thank you for your thoughtful suggestion. The p-values in Supplementary Table 13 were corrected for multiple tests in our revised manuscript. As we only tested for second level GO terms here, thus multiple correction was carried out for the 54 second level GO terms, after correction, our result did not change quite a lot. GO categories still significant were marked with red color. The description in the revised manuscript was changed to:

“For example, almost all second level Gene Ontology (GO) categories had fewer genes in *G. elata* than in the other two species, and 9 of these categories (16.7%) were significantly reduced (Fisher’s Exact test, $p < 0.05$, Supplementary Fig. 6 and Supplementary Table 13).”

(3) Re:

“Although the *G. elata* genome has clearly undergone extensive gene loss, we found 197 that 430 gene families (19 by a significant margin), containing 1532 genes (184 by a significant margin), showed expansion in *G. elata* compared to *P. equestris*, *D. 199 officinale*, and *A. comosus* (Supplementary Fig. 7 and Supplementary Table 32-33). “

— similarly, the authors must correct for multiple tests and modify conclusions accordingly

Thank you for your reminder. Expanded gene families were detected through Café program, the program uses a birth and death process to model gene gain and loss over a phylogeny. P-values in this program were calculated by random sampling methods, thus no multiple correction was required for Café results (Supplementary Fig. 7).

While p-values presented in all tables related GO enrichment analysis in our previous manuscript were indeed corrected for multiple test through false discovery rate methods (Supplementary Table 16 and 32), we had marked this at the end of the table (*Fisher’s exact test, corrected by false discovery rate). To avoid confusions, in our revised manuscript, we have modified “p-values” into “adjusted p-values” in this table.

As suggested, we change the p-values in Supplementary Table 14 and 33 into adjusted p-values in our revised manuscripts. After correction, only two Pfam families remain significant. However, this change did not affect our description in the main text:

“Although the *G. elata* genome has clearly undergone extensive gene loss, we found

that 430 gene families (19 by a significant margin), containing 1532 genes (184 by a significant margin), showed expansion in *G. elata* compared to *P. equestris*, *D. officinale*, and *A. comosus* (Supplementary Fig. 7 and Supplementary Table 32-33).”

(4) The authors and editors MUST search through the ENTIRE TEXT for cases where statistical results may have been presented without correction for multiple tests, as in (2) and (3) above. It is never satisfactory to avoid this when so many GOs (for example) are being tested for enrichment all at the same time.

Thank you for your thoughtful suggestion. We carefully check all supplementary tables. Then we changed p-values into adjusted p-values for Supplementary Table 13, Supplementary Table 14 and Supplementary Table 33.

While p-values presented in all tables related GO enrichment analysis were indeed corrected for multiple test through false discovery rate methods. To avoid confusions, in our revised manuscript, we have modified “p-values” into “adjusted p-values” in all these tables.

For tables that listing number of copies for specific genes, like Supplementary Table 27, Supplementary Table 38, Supplementary Table 40 and Supplementary Table 41, we did not do multiple corrections. The reason is that tests for each gene can be considered as independent experiments. Beside statistical tests, the gain/loss of specific genes were highlighted by phylogenetic trees.

(5) You must not phrase the text exactly as the reviewer noted:

“(4) Actually, *G. elata* do have reduced leaves (=bracts) – take a look at pictures of the scape --- although field guides and orchid systematists like to pretend they are leafless! The reviewer is correct. We have change these statements as 'Gastrodia elata (Orchidaceae) is an orchid popularly used in traditional Chinese medicine that has a fully mycoheterotrophic lifestyle with a 'leafless' and rootless growth habit (actually they have highly reduced leaves, bracts in scape, although field guides and orchid systematists like to pretend they are leafless).”

— you could say “...although field guides and systematists often refer to the plants as leafless.”

Thank you for your suggestion. We revised it as '*Gastrodia elata* (Orchidaceae) is an orchid popularly used in traditional Chinese medicine that has a fully mycoheterotrophic lifestyle with highly reduced leaves and bracts in scape, although field guides and systematists often refer to the plants as leafless.’